

# BGC-val: a model and grid independent python toolkit to evaluate marine biogeochemical models

Lee de Mora[1], Andrew Yool[2], Julien Palmieri[2], Alistair Sellar[3], Till Kuhlbrodt[4], Ekaterina Popova[2], Colin Jones[5], and J. Icarus Allen[1]

[1]Plymouth Marine Laboratory, Prospect Place, The Hoe, Plymouth, PL1 3DH, UK
[2]National Oceanography Centre, University of Southampton Waterfront Campus, European Way, Southampton SO14 3ZH, UK
[3]Met Office Hadley Centre, Exeter, EX1 3PB, UK
[4]NCAS, Department of Meteorology, University of Reading, Reading, RG6 6AH, UK
[5]NCAS, School of Earth and Environment, University of Leeds, Leeds, LS2 9JT, UK

*Correspondence to:* Lee de Mora (ledm@pml.ac.uk)

**Abstract.** The biogeochemical evaluation toolkit, BGC-val, is a model and grid independent python toolkit that has been built to evaluate marine biogeochemical models using a simple interface. Here, we present the ideas that motivated the development of the BGC-val software framework, introduce the code structure, and show some applications of the toolkit using model results from the Fifth Climate Model Intercomparison Project (CMIP5).

The key ideas that directed the toolkit design were model and grid independence, front loading analysis functions and regional masking, interuptabililty, and ease of use. We present each of these goals, why they were important and what we did to address them. We also present an outline of the code structure of the toolkit, illustrated with example plots produced by the toolkit.

After describing BGC-val, we use the toolkit to investigate the performance of the marine circulate and biogeochemical parts
of the CMIP5 models, and highlight some predictions about the future state of the marine ecosystem under a business as usual $CO_2$ concentration scenario (RCP 8.5).

## 1   Introduction

It is widely known that climate change is expected to have a significant impact on weather patterns, the cryosphere, the land surface and the ocean (Stocker et al., 2015; Cook et al., 2013; Le Quéré et al., 2013; Rhein et al., 2013). Marine organ-
isms are vulnerable not only to impacts of rising temperatures, but also to associated deoxygenation (Stramma et al., 2008; Gruber, 2011), as well ocean acidification driven by ocean $CO_2$ uptake (Caldeira and Wickett, 2003; Dutkiewicz et al., 2015; Azevedo et al., 2015).

The 2016 Paris Climate Accord is a wide ranging international agreement on greenhouse gas emissions mitigation and climate change adaptation which is underpinned by the goal of limiting the global mean temperature increase to less than 2° C
above pre-industrial levels, (Schleussner et al., 2016). International environmental policies like the Paris Climate Accord hinge on the projections made by the scientific community. Numerical models of the Earth system are the only tools available to





make meaningful predictions about the future of our climate. However, in order to trust the results of the models, they must first be demonstrated to be a sufficiently good representation of the Earth system. The process of testing the behaviour of the simulations is known as model evaluation. The importance of evaluating the models grows in significance as models are increasingly used to inform policy (Brown and Caldeira, 2017).

The Coupled Model Intercomparison Project (CMIP) is a framework for coordinating climate change experiments and to provide information of value to the International Panel on Climate Change (IPCC) Working Groups, (Taylor et al., 2012). CMIP5 was set up to address outstanding scientific questions, to improve understanding of climate, and to provide estimates of future climate change that will be useful to those considering its possible consequences (Taylor et al., 2007; Meehl et al., 2009). These models represent the best scientific projections of the range of possible climates going into the 21st century. The

results of previous rounds of CMIP comparisons have become a crucial component of the IPCC reports. In the fifth phase of CMIP, many of the climate forecasts were based on representative concentration pathways (RCP), which represented different possibilities for greenhouse gases concentrations in the 21st century (Moss et al., 2010; van Vuuren et al., 2011).

The upcoming sixth climate model intercomparison project, CMIP6, (Eyring et al., 2016), is expected to start receiving models in the year 2018. In order to contribute to CMIP6, each model must complete suite of scenarios know as the Diagnosis,

Evaluation, and Characterisation of Klima (DECK) simulations, which include an atmospheric model intercomparison between the years 1979 to 2014, a pre-industrial control simulation, a 1% per year $CO_2$ increase, an abrupt $4 \times CO_2$ run, and a historical simulation using CMIP6 forcings (1850-2014). CMIP6 models are also required to use consistent standardisation, coordination, infrastructure, and documentation.

One of the primary goals of the CMIP5 and CMIP6 is to provide meaningful projections on the ultimate fate of the carbon

emitted by humanity into the atmosphere. The ocean is an important sequesterer of carbon, absorbing approximately 27% of the anthropogenic carbon emitted between 2002 and 2011 (Le Quéré et al., 2013). Under a changing climate, the ocean is likely to continue to absorb some of the anthropogenic atmospheric carbon dioxide, rendering the ocean more acidic via the increased formation of carbonic acid. The acidification of the ocean is expected to continue to have significant impact on sea life (Dutkiewicz et al., 2015; Rhein et al., 2013).

Due to the high thermal capacity of water, nearly all of the excess heat captured by the greenhouse effect is absorbed by the ocean (Rhein et al., 2013). This increases the temperature of the waters, which causes sea levels to rise via thermal expansion (Church et al., 2013), may accelerate the melting of sea ice, (Moore et al., 2015), and may push many marine organisms outside of their thermal tolerance range, (Poloczanska et al., 2016).

Numerical simulations are the only tools available to predict how these factors and others will influence marine life under

a changing climate. Furthermore, Earth system models are also the tool that can project how changes in the marine system feedback on, and interact with other climate-relevant components of the Earth system. The UK Earth System Model (UKESM1) is a next generation Earth system model currently under development. The aim of UKESM1 is to develop and apply a world-leading Earth System Model. Simulations made with the UKESM1 will be contributed to CMIP6.

During the process of building the UKESM1, we also deployed a suite of tools to monitor the marine component of the

model as it was being developed. This software suite is called BGC-val, and it is used to compare the marine component



of simulations against each other and against observational data, with an emphasis on marine biogeochemistry. The suite of evaluation tools that we present in this work is a generalised extension of those tools. The focus of this work is not to prepare a guide on how to run the BGC-val code, but rather to present the central ideas and methods used to design the toolkit. Details on how to install, set up and run the code can be found in the appendix A, and more detail is available in the `README.md` file
in the repository.

After this introduction, sect. 2 outlines the features of the BGC-val Toolkit, sect. 3 describes the evaluation process used by the toolkit, and sect. 4 describes the code structure of the BGC-val toolkit. Finally, sect. 5 shows some examples of the toolkit in use with model data from CMIP5.

### 1.1   The development and evaluation of marine models

The evaluation of marine ecosystem models is a crucial stage in the deployment of climate models to inform policy decisions. When compared to models of other parts of the Earth system, marine models have several unique features which complicate the model development and evaluation process.

To preface these comments, it is important to remember than any component model of an Earth system model can always be expanded to include more complexity, (Washington et al., 2009). However, the additional complexity may push the
model beyond what is currently computationally feasible (Kwiatkowski et al., 2014) or what can be supported by observations (Anderson, 2005; Flynn, 2006). Furthermore, adding model complexity will typically increase model costs, or decrease model running speed. This can result in project delays, increase running costs or limit the range of scenarios than can be investigated. There are many trade offs involved and model developers need to make pragmatic design decisions. For this reason, the comparisons highlighted in this section are for illustrative purposes only.

When compared to similarly complex models of the land surface, model of the ocean have significantly higher computational costs, (Balaji et al., 2017). Land surface models are typically two dimensional (Best et al., 2011; Clark et al., 2011; Best et al., 2015), whereas oceans models are almost always three dimensional. In addition, while the land surface does change over time, it is not subjected to movement in the same way that sea life occupies an environment of flowing water. The circulation of the ocean has a significant impact on local conditions. For instance, the strength of the surface mixing influences the availability
of nutrients for phytoplankton. The need for full three dimension models and the advection of each tracer field of the marine model are both associated with a significant additional computational burden. While the physical system drives the biology, the biology can also influence physical circulation (Manizza et al., 2005; Kunze, 2006). As such, the circulation needs to be developed, evaluated, and tuned alongside the biogeochemistry.

Due to the shorter times scales and the larger number of advected tracers, models of the atmosphere can be more computa-
tionally expensive than models of the ocean for each simulated year (Balaji et al., 2017). However, more simulated years are needed for the ocean models to become equilibrated than for atmospheric models. In this case, we define the equilibrium as where the model is in a steady-state, meaning that the temporal derivatives of tracer fields are close to zero. In some parts of the ocean, once water is subducted into the deep ocean, it may take more than a thousand years to come in contact with the atmosphere again (Stuiver et al., 1983; Gebbie and Huybers, 2012). To become fully equilibrated (also known as "spun up"),



the ocean must repeat several such mixing cycles (Séférian et al., 2016). In contrast, the mixing timescales observed in the atmosphere is measured in days, (Good et al., 2003; Ehhalt and Prather, 2001). This means that the simulated time needed to bring a model to equilibrium is significantly higher for an ocean-only model than an atmosphere-only model.

The data available for evaluating a marine model can also be relatively scarce. The ocean covers more than twice as much
of the surface of the Earth than land, and there are sizeable regions of the ocean which are rarely visited by humans, let alone sampled by scientific cruises (Garcia-Castellanos and Lombardo, 2007). In addition, only the surface of the ocean is visible to satellites; the properties of marine life in the deep waters can not be inferred from remote sensing. Similarly, the connections between different components of the Earth system can also be difficult to measure. Several crucial global fluxes are unknown or estimated with significant uncertainties, such as the global total flux of $CO_2$ into the ocean, (Takahashi et al., 2009), or the
global total deposition of atmospheric dust, (Mahowald et al., 2005).

When developing a model of the ocean, the early stages of tuning and development are done under pre-industrial atmospheric $CO_2$ concentrations. However, it is not ideal to compare a pre-industrial model against modern measurements of the ocean. Furthermore, it is not possible to compare the spin up against pre-industrial data, because there is few pre-industrial data. This means that the tuning stages is often done by comparing a pre-industrial model field against a recent measurement dataset; this
is not ideal yet unavoidable.

In summary, the slow overturning and ventilation of the ocean means that the modelled marine environment requires significantly longer duration simulations to equilibrate than other climate-relevant Earth system. Furthermore, there is a scarcity of both pre-industrial and modern observational data which can be used to compare the systems behaviour against nature. This means that the process of iteratively developing and tuning the marine component of an Earth system model can be slow and
computationally expensive. The BGC-val toolkit was developed to assist with evaluating this process.

## 1.2 Model evaluation using ESMvalTool

The evaluation of marine ecosystem models is a crucial part of the development and deployment of simulations of the marine environment. However, there is no standardised toolkit specific to this field and evaluation is typically performed in an ad hoc manner. BGC-val provides a framework to compare model and observational data on a global, regional or local scale, and
summarise the model's behaviour in a clear, shareable user-friendly interface.

As part of preparation of CMIP6, a community diagnostic toolkit for evaluating climate models, called ESMvalTool is being developed, (Poloczanska et al., 2016). There are many similarities between BGC-val and ESMvalTool. ESMvalTool is also a flexible evaluation toolkit and many of the features developed for BGC-val also appear in ESMvalTool. However, BGC-val was developed explicitly for evaluating models of the ocean, whereas ESMvalTool was built to evaluate models of the entire
Earth system.

BGC-val has been deployed operationally since June 2016, where it has been used extensively for the development, evaluation and tuning of the spin up of the marine component of the UKESM1, MEDUSA,(Yool et al., 2013). It must be noted that ESMvalTool was not yet available for operational deployment at that point. Furthermore, many of the features that were implemented into BGC-val have since also been added to ESMvalTool, but were not available at that time. In addition, many



of the metrics deployed in BGC-val's ocean-specific evaluation have been proposed as key metrics to include in future versions of ESMvalTool.

Ocean Assess is an UK Met Office software toolkit for evaluating the physical circulation of the models developed there. Ocean Assess is not available outside the met office and is not yet described in any public facing documentation.

## 2   The BGC-val Toolkit design features

While BGC-val was originally built as a toolkit for investigating the time development of the marine biogeochemistry component of the UK Earth system model, UKESM1, the primary focus of the BGC-val's development was to make the toolkits as generic as possible. This means that the tools can be easily adapted for use with for a wide range of models, spatial domains, model grids, fields, datasets and time scales without needing significant changes to the underlying software, and without any significant post-processing of the model or observational data. The toolkit was built to be model independent, grid independent, interuptable, simple to use, and to include front loading analyses and masking functionality.

The BGC-val toolkit was written in python 2.7. The reason that python was used is because it is freely available and widely distributed, it is portable and available with most operating systems, there are many powerful standard packages can be easily imported or installed locally, it is object oriented (allowing front loading functionality described below), and it is popular and hence well documented and well supported.

### 2.1   Model independence

The Earth system models submitted to CMIP5 were created by largely independent groups of scientists. While some model developers build CMIP compliance into their models, other model developers choose to reformat the file names and contents to a uniform naming and units scheme before they are submitted to CMIP. This flexibility means that each model working group may use their own file naming conventions, dimension names, variables and variable names until the data is submitted to CMIP5. Outside of the CMIP5 standardisation, there are many competing nomenclatures. For instance, in addition to the CMIP standard name, `lat`, we have encountered the following non-standard names in model data files, all describing the latitude coordinate: `lats`, `rlat`, `nav_lat`, `latitude` and several other variants. Similarly, different models and observational datasets may not necessarily use the same units.

While the CMIP5 data has been produced using a uniform naming scheme, this toolkit allows for models to be evaluated without any prior assumptions on their naming conventions or units. This means that it would be possible to deploy this toolkit during the development stage of a model, before reformatting the data to CMIP compliance. This is how this toolkit was applied during the development of UKESM1. Model independent ensures that the toolkit can be applied in a range of scenarios, without requiring significant knowledge of the toolkit's inner workings, and without post-processing the data.



## 2.2 Grid independence

An important part of producing a model of the Earth system is to divide the ocean into a grid of small boxes, each of which represents the mean behaviour of the ocean in small region. The descriptions of the boundaries between these boxes is known as the model grid and the boxes are called grid cells. Each Earth system model submitted to CMIP5 chose how they wanted

to divide the ocean. Furthermore, unlike the naming and unit schemes, the model data submitted to CMIP5 has not been reformatted to a uniform grid.

The BGC-val toolkit was originally built to work with NEMOs extended eORCA1 grid, which is a tri-polar grid with an irregular distribution of two dimensional latitude and longitude coordinates. However, information about the grid is supplied alongside the model data, such that there is no grid requirement hard-wired into BGC-val. This means that the toolkit is capable

of handling any kind of model grid without the need to re-interpolate the data to a common grid.

Making the toolkit grid independent means that the toolkit can be applied without any hard-wired grid descriptions in the toolkit. As such, the toolkit will function in the same way regardless of whether the model uses a regular grid, reversed grid, a tripolar grid, or any other type of grid.

When calculating means, medians and other metrics, the toolkit uses the grid cell area or volume to weight the results. This

means that it is possible to use this toolkit to compare multiple models that use different grids without the computationally expensive and potentially lossy process of re-interpolation to a common grid. The CMIP5 datasets include grid cell area and volume. However, outside the CMIP standardised datasets, most models and observational datasets provide grid cell boundaries or corner coordinates as well as longitude and latitude cell centred coordinates. These corners and boundaries can be used to calculate the area and volume of each grid cell. If only the cell centered points are provided, the BGC-val toolkit is able to

estimate the grid cell area and volume based on the coordinates.

## 2.3 Front loaded analysis functionality

While extracting the data from file, BGC-val can apply an arbitrary pre-defined or user-defined mathematical python function to the data. This means that it is straightforward to define a customised analysis function in a python script, then to pass that function to the evaluation code, which then applies the analysis function to the dataset as the data is loaded. Firstly, this method

ensures that the toolkit is not limited to a small set of pre-defined functions. Secondly, the end users are not required to go deep into the code repository in order to use a customised analysis function.

In its simplest form, the front loading functionality allows a straightforward conversion of the data as it is loaded. As a basic example, it would be straightforward to add a function to convert the Temperature fields units from Celsius to Kelvin. The Celsius to Kelvin function would be written in a short python script, the script would be listed by name in the evaluation's

configuration file. This custom function would be applied while loading the data, without requiring the model data to be pre-processed, or for the BGC-val inner workings to be edited in depth.

Similarly, more complex analysis functions can also be front loaded in the same way. For instance, the calculation of the global total volume of oxygen minimum zones, or the total flux of $CO_2$ from the atmosphere to the ocean, or the calculation



of the total current passing though the Drake Passage, are all relatively complex calculations which can be applied to datasets. These functions are also already included in the toolkit, in the `functions` folder, described in sect. 4.3.1.

## 2.4 Regional masking

Similarly to the front loading analyses described above in sect. 2.3, BGC-val users can pre-define a customised region of interest, then ignore data from outside that region. The regional definitions is supplied in advance and can be used to evaluate several models or datasets. The process of hiding data from regions that are not under investigation is know as "masking". In addition, while the UKESM and other CMIP model are global models, there is no requirement for the model to be global in scale; regional and local models can also be investigated using BGC-val.

While BGC-val already includes many regional masks, it is straightforward to define new to that hide regions which are not under investigation. Similarly to section 2.3, the new masks can be defined in an advance, named, and called by name, without having to go deep into the toolkit code. These masks can be defined in terms of the latitude, longitude, depth, time range, or even the data itself. The toolkit include several standard masks, for example there is a mask which allows the user to retrieve only data in the Northern Hemisphere, called `NorthernHemisphere`, or to ignore all data deeper than 10 m, called `Depth_0_10m`.

However, more complex masks could be created. For instance, it is feasible to make a custom mask which ignores data below the $5^{th}$ percentile or above the $95^{th}$ percentile. It is also possible to stack masks, by applying two or more masks successively in a custom mask. For instance, a hypothetical custom mask could mask data below a depth of 100 m, ignoring the Southern Ocean and also remove all negative values. This means that it is straightforward for users to add arbitrarily complex regional masks to the dataset. For more details, please see sect. 4.3.2.

## 2.5 Interuptable

BGC-val makes regular save points during data processing, such that the analysis can be interrupted and restarted without reprocessing all the data files from the beginning. This means that each analysis only needs to run once. Alternatively, it means that it is possible to evaluate on-going model simulations, without reprocessing everything every time that the evaluation is needed.

The processed data are saved as python shelve files. Shelve files allows for any python object, including data arrays and dictionaries to be committed to disk. As the name suggests, shelving allows for python objects to be stored and reloaded at a later stage. These shelve files help with comparison of multiple models or regions, as the evaluation results can be set aside then quickly reloaded later to be processed into a summary figure, or pushed into a human readable data file.

## 2.6 Ease of use

A key goal was to make the toolkit straightforward to access, install, set up, and use. The code is accessed using a gitlab server, which is a private online graphical user interface to the version control software, git, similar to the commercial GitHub service.



This makes it straightforward for multiple users to download the code, report bugs, develop new features and share the changes. While this is a private gitlab server, and users can register for access here: http://www.pml.ac.uk/Modelling_at_PML/Access_Code. BGC-val behaves like a standard python package, and can be installed via the "pip" interface.

More importantly, BGC-val was built such that entire evaluation suites can be run from a single human readable configura-
tion file. This configuration file uses the .ini configuration format, and doesn't require any knowledge of python or the inner workings of BGC-val. The configuration file contains all the paths to data, descriptions of the data file and model data, links to the evaluation function, Boolean switches to turn on and off various evaluation metrics, the names of the variables needed to perform an evaluation, as well as the paths for the output files. This makes it possible to run the entire package without having to change more than a single file. The configuration file is described in section 4.1.

BGC-val also summarises the results into an html document, which can be opened directly in a web browser, and evaluation figures can be extracted for publication or sharing. The summary report is described in section 4.2.3. Two examples of the summary report are included in the supplementary documents.

## 3   Evaluation process

In this section, we describe the five stage evaluation process that the toolkit applies to model and observational data. Figure 1
summarises the evaluation process graphically.

### 3.1   Load model and observational data

The first stage of the evaluation process is to load the model and observational data. The model data is typically a time series of two or three dimensional variables stored in a one or several NetCDF files. Please note that we use the standard convention of only counting spatial dimensions. As such, any mention of dimensionality here implies an additional temporal dimension,
ie: three dimensional model data has length, height, width and time dimensions.

The model data can be a single NetCDF variable, or some combination of several variables. For instance, in some marine biogeochemistry models, total chlorophyll concentration is calculated as the sum of many individual phytoplankton functional type chlorophyll concentrations. In all cases, BGC-val loads the model data one time step at the time, whether the NetCDFs contains one or multiple time steps.

The front loading evaluation functions described in sections 2.3 and 4.3.1 are applied to each time step of the model data at this point. The resulting loaded data can be a one, two or three dimensional array. The use of an observational dataset is optional, but allows the model to be compared against historical measurements. The observational data and model data are not required to be loaded using the same function.

When loading data, BGC-val assumes that we use the NetCDF format. The NetCDF files are opened in BGC-val with
a custom python interface, `dataset.py` in the `bgcvaltools` package. The `dataset` class is based on the standard python `netCDF4.Dataset` class. NetCDF files are composed of two parts, the header and the data. The header typically includes all the information needed to understand the origin of the file, while the data contains a series of named variables.



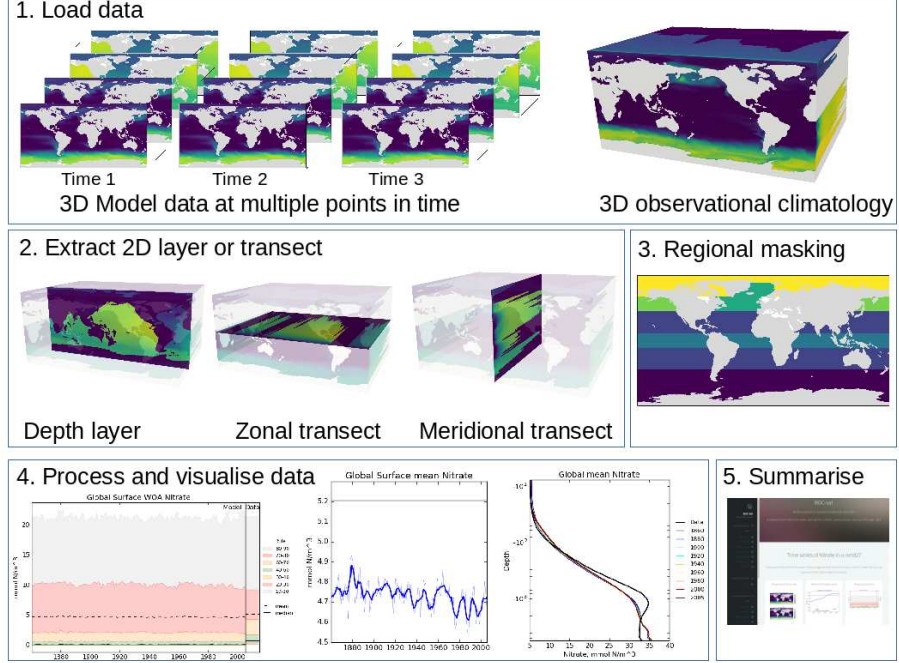

**Figure 1.** The five stages of the evaluation process. The first stage is the loading of the model and observational data. The second stage is the extraction of a two dimensional array. The third stage is regional masking. The fourth stage is processing and visualisation of data. The fifth stage is the publication of an html summary report. Note that the three figures shown in the fourth stage are repeated below in figs. 3, 4, and 6.

Each named variable should (but not obligatorily) include their dimensions, units, their long name, their data and their mask. Furthermore, the dimensions in NetCDF format are not restricted to regular latitude-longitude grids. Some NetCDFs use arbitrary dimensions, such as a grid cell index, irregular grids like NEMOs eORCA1 grid, or even triangular grid cells as in the Finite Volume Coastal Ocean Model (FVCOM) (Chen et al., 2006).

## 3.2   Extract a two dimensional slice

The second evaluation stage is the extraction of a two dimensional variable from three dimensional data. As shown in the second pane of Fig. 1, the two dimensional variable can be the surface of the ocean, a depth layer parallel to the surface, an East-West transect parallel to the equator, or a North South transect perpendicular to the equator. This stage is included in order to speed up the process of evaluating a model; in general, it is much quicker to evaluation a 2D field than a 3D field. Furthermore, the spatial and transect maps produced by the evaluation process can become visually confusing when overlaying several layers. Note that stages 2 and 3 are applied to both model and observational data (if present). This stage is unnecessary if the data loaded in the first stage is already a two dimensional variable, such as the fractional sea ice coverage, or a one dimensional variable, such as the Drake Passage Current.





In the case of a transect, instead of extracting along the files internal grid, the transect is produced according to the geographic coordinates of the grid. This is done by locating points along the transect line inside the grid cells, based on the grid cell corners.

### 3.3 Extract a specific region

Stage three is the masking of specific regions or depth levels from the 2D extracted layer, as described in section 2.4. Stage three is not needed if the variable is already a one dimensional product, such as the total global flux of $CO_2$. Stage three takes the two dimensional slice, then converts the data into five one dimensional arrays of equal length. These arrays represent the time, depth, latitude, longitude and value of each data point in the data. These five 1D arrays can be further reduced by making cuts based on any of the coordinates, or even cutting according to the data itself.

Both stages two and three of the evaluation process reduce the number of grid cells under evaluation. This two-stage process is needed because the stage three masking cut can become memory intensive. As such, it is best to for the data to arrive at this stage in a reduced format. In contract, the stage two process of producing a 2D slice is a relatively computationally cheap process. This means that the overall evaluation of a model run can be done much faster.

### 3.4 Produce visualisations

Stage four is the processing of the two dimensional datasets and the creation of visualisations of the model and observational data. Figure 1 shows three examples of the visualisations that BGC-val can produce the time series of the spread of the data, a simple time series, the time development of the depth profile. However, several other visualisations can also be produced: for instance, the point to point comparisons of model data against observational data, and comparison of the same measurement between different regions, times or model, or scenarios.

Which visualisations are produced depends on which evaluation switches are turned on, but also a range of other factors including the dimensionality of the model dataset, and the presence of an observational dataset. For instance, figures that show the time development of the depth profile require three dimensional data. Similarly, the point to point comparison requires an observational dataset to for the model to match against. More details on the range of plotting tools are available in section 4.2.1, and 4.2.2.

Stages one to four are repeated for each evaluated field, for multiple models, scenarios or different versions of the same model. If multiple jobs or models are requested, then comparison figures can also be created in stage four.

### 3.5 Produce a summary report

The fifth stage is the automated generation of a summary report. This is an html document which shows the figures that were produced as part of stages 1-4. This document is built from html and can be hosted and shared on a web server. More details on the report are available in section 4.2.3.





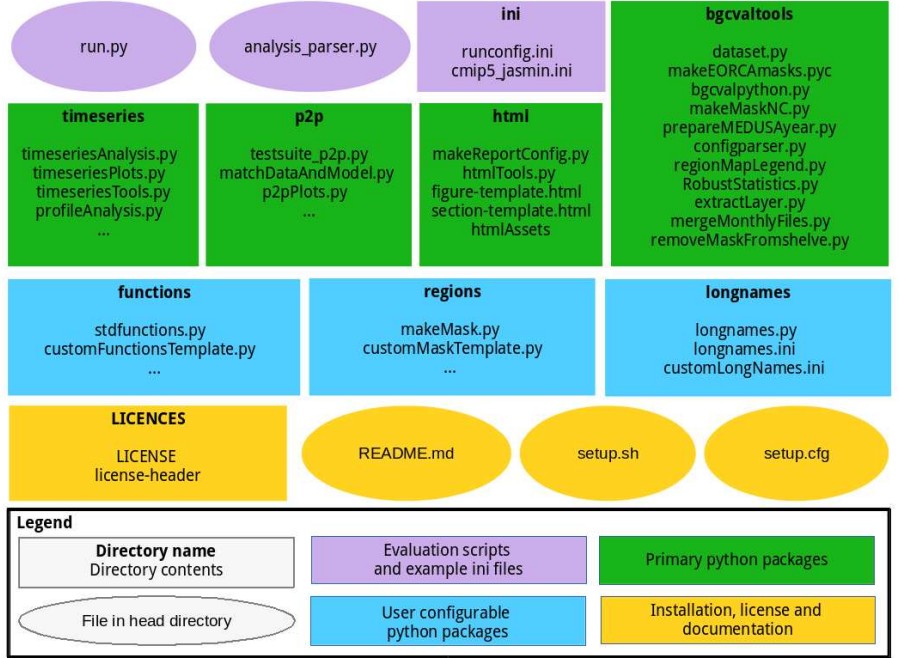

**Figure 2.** The structure of the BGC-val repository. The principal directories are shown as rectangles, with the name of the directory in bold followed by the key files contained in that directory. Individual files in the head directory are shown with rounded corners. The evaluation scripts and the configuration directory are shown in purple. The primary python modules are split into four directories, shown in green rectangles. The three user configurable python modules are shown as blue rectangles. The license, read-me and set up files are shown in yellow.

## 4  Code structure and functionality

The directory structure of the BGC-val toolkit repository is summarised in fig. 2. This figure highlights a handful of the key directories and files of the toolkit. We use the standard python nomenclature, where applicable. In python, a module is a python source file, which can expose classes, functions and global variables. A package is simply a directory containing one or more modules and python creates a package using the `__init__.py` file. The BGC-val toolkit contains seven packages and dozens of modules.

In this figure, ovals are used to show single files in the head directory, and rectangles show folders or packages. In the top row of fig. 2, the are two purple ovals and a rectangle which represent the important evaluation scripts and the example configuration files. These files include the `run.py` executable script which is a user-friendly wrapper for the `analysis_parser.py` script, (also in the head directory). The `analysis_parser.py` file, is the principal python file that loads the run configuration and launches the individual analyses. The `ini` directory includes several example configuration files, including the





configuration files that were used to produce the figure in this document. Note that the `ini` directory is not a python package, just a repository that holds several files. A full description of the functionality of configuration files can be found in sect. 4.1.

The four main python packages in BGC-val are shown in green rectangles in fig. 2. Each of these modules has a specific purpose: the `timeseries` package described in sect. 4.2.1 performs the evaluation of the time development of the model,

the `p2p` package described in sect. 4.2.2 does an in-depth spatial comparison of a single point in time for the model against a historical data field, the `html` package described in sect. 4.2.3 contains all the python functions and html templates needed to produce the html summary report. This `bgcvaltools` package contains many python routines that perform a range of important functions in the toolkit. These tools include, but are not limited to: a tool to read NetCDF files, a tool to extract a specific 2D layer or transect, a tool to read and understand the configuration file, and many others.

The three user configurable packages `functions`, `regions` and `longnames` are shown in blue in fig. 2. The `functions` package, described in sect. 4.3.1, contains all the front loading analysis functions described in sect. 2.3, which are applied in the first stage of the evaluation process described in sect. 3. The `regions` package described in sect. 4.3.2 contains all the masking tools described in sect. 2.4, and which are applied in the third stage of the evaluation process described in sect. 3. The `longnames` package , described in sect. 4.3.3, is a simple tool which behaves like a look up dictionary, allowing users to link

human readable or "pretty" names (like "Chlorophyll") against the internal code names or shorthand (like "chl"). The pretty names are used in several places, notably in the figure titles, legends and on the html report.

The `licenses` directory, the setup configuration files and the `README.md` are all in the main directory of the folder, shown in yellow in fig. 2. The licenses directory contains information about the Revised Berkeley Software Distribution (BSD) 3-clause license. The `README.md` file contains specific details on how to install, set up and run the code. The `setup.py` and

`setup.cfg` files are used to install the BGC-val toolkit.

## 4.1 The Configuration file

The configuration file is central to the running of BGC-val and contains all the details needed to evaluate a simulation. This includes the file path of the input model files, the users choice of analysis regions, layers and functions, the names of the dimensions in the model and observational files, the final output paths, and many other settings. All settings and configuration

choices are recorded in an single file, using the `.ini` format. Several example configuration files can also be found in the `ini` directory. Each BGC-val configuration file is composed of three parts: an Active keys section, a list of evaluation sections, and a Global section. Each of these parts are described below.

The tools that parse the configuration file is in the `configparser.py` module in the `bgcvaltools` package. These tools interpret the configuration file and use them to direct the evaluation. Please note that we use the standard `.ini` format

nomenclature while describing configuration files. In this, `[Sections]` are denoted with square brackets, each option is separated from its value by a colon, ":", and the semi-colon ";" is the comment syntax in `.ini` format.



### 4.1.1 Active keys section

The active keys section should be the first section of any BGC-val configuration file. This section consists solely of a list of Boolean switches, one Boolean for each field that the user wants to evaluate:

```
   [ActiveKeys]
Chlorophyll      : True
   A                : False
   ; B              : True
```

To reiterate the `ini` nomenclature, in this example `ActiveKeys` is the section name, and `Chlorophyll`, `A`, and `B` are options. The values associated with these options are the Boolean's, `True`, `False`, and `True`. The option `B` is commented

out and will be ignored by BGC-val.

In the `[ActiveKeys]` section, only options whose values are set to `True` are active. False Boolean's values and commented lines are not evaluated by BGC-val. In this example, the `Chlorophyll` evaluation is active, but both options `A` and `B` are switched off.

### 4.1.2 Individual evaluation sections

Each `True` Boolean options in the `[ActiveKeys]` section needs an associated `[Section]` with the same name as the option in `[ActiveKeys]` section. The following is an example of an evaluation section for chlorophyll in the HadGEM2-ES model.

```
   [Chlorophyll]
   name             : Chlorophyll
units            : mg C/m^3

   ; The model name and paths
   model            : HadGEM2-ES
   modelFiles       : /Path/*.nc
modelgrid        : CMIP5-HadGEM2-ES
   gridFile         : /Path/grid_file.nc

   ; Model coordinates/dimension names
model_t          : time
   model_cal        : auto
   model_z          : lev
```



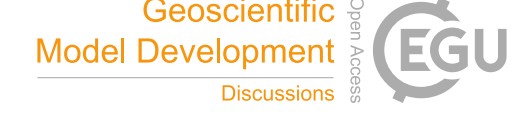

```
model_lat        : lat
model_lon        : lon

; Data and conversion
5  model_vars       : chl
model_convert    : multiplyBy
model_convert_factor : 1e6
dimensions       : 3

; Layers and Regions
layers               : Surface 100m
regions              : Global SouthernOcean
```

The `name` and `units` options are descriptive only; they are shown on the figures and in the html report, but do not influence
the calculations. This is set up so that the name associated with the analysis may be different to the name of the fields being
loaded. Similarly, while NetCDF files often have units associated with each field, they may not match the units after the user
has applied an evaluation function. For this reason, the final units after any transformation must be supplied by the user. In
the example showed here, HadGEM2-ES correctly used the CMIP5 standard units for chlorophyll concentration, kg m$^{-3}$.
However, we prefer to view Chlorophyll in units of mg m$^{-3}$.

The `model` option is typically set in the `Global` section, described below in sect. 4.1.3, but it can be set here as well. The
`modelFiles` option is the path that BGC-val should use to locate the model data files on local storage. The `modelFiles`
option can point directly at a single NetCDF file, or can point to many files using wild-cards (`*`, `?`). The file finder uses the
standard python package, `glob`, so wild-cards must be compatible with that package. Additional nuances can be added to the
file path parser using the placeholders `$MODEL`, `$SCENARIO`, `$JOBID`, `$NAME` and `$USERNAME`. These placeholders are
replaced with the appropriate global setting they are read by the `configparser` package. The global settings are described
below in section 4.1.3. For instance, if the configuration file is set to iterate over several models, then the `$MODEL` placeholder
will be replaced by the model name currently being evaluated.

The `gridFile` option allows BGC-val to locate the grid description file. The grid description file is a crucial requirement
for BGC-val, as it provides important data about the model mask, the grid cell area, the grid cell volume. Minimally, the
grid file should be a NetCDF which contains the following information about the model grid: the cell centred coordinates for
longitude, latitude and depth, and these fields should use the same coordinate system as the field currently being evaluated. In
addition, the land mask should be recorded in the grid description NetCDF in a field called `tmask`, the cell area should be in a
field called `area` and the cell volume should be recorded in a field labelled `pvol`. BGC-val includes the `meshgridmaker`
module in the `bgcvaltools` package and the function `makeGridFile` from that module can be used to produce a grid




file. The `meshgridmaker` module can also be used to calculate the cross sectional area of an ocean transect, which is used in several flux metrics such as the Drake passage current or the Atlantic Meridianal Overturning circulation.

Certain models use more than one grid to describe the ocean; for instance NEMO uses a U grid, a V grid, a W grid, and a T grid. In that case, care needs to be taken to ensure that the grid file provided matches the data. The name of the grid can be set with the `modelgrid` option.

The names of the coordinate fields in the NetCDF need to be provided here. They are `model_t` for the time, `model_cal` for the model calendar. Any NetCDF calendar option (360_day, 365_day, standard, Gregorian, etc ...) is also available using the `model_cal` option, however, the code will preferentially use the calendar included in standard NetCDF files. For more details, see the `num2date` function of the `netCDF4` python package, (https://unidata.github.io/netcdf4-python/). The depth, latitude and longitude field names are passed to BGC-val via the `model_z`, `model_lat` and `model_lon` options.

The `model_vars` option tells BGC-val the names of the model fields that we are interested in. In this example, the CMIP5 HadGEM2-ES chlorophyll field is stored in the NetCDF under the field name `chl`. As mentioned already, HadGEM2-ES used the CMIP5 standard units for chlorophyll concentration, kg m$^{-3}$, but we prefer to view Chlorophyll in units of mg m$^{-3}$. As such, we load the chlorophyll field using the conversion function, `multiplyBy` and give it the argument 1e6 with the `model_convert_factor` option. More details are available below in sect. 4.3.1 and in the `README.md` file.

BGC-val uses the coordinates provided here to extract the layers requested in the `layers` option from the data loaded by the function in the `model_convert` option. In this example that would be the surface and the 100 m depth layer. For the time timeseries and profile analyses, the layer slicing is applied in the `DataLoader` class in the `timeseriesTools` module of the `timeseries` package. For the point to point analyses, the layer slicing is applied in the `matchDataAndModel` class in the `matchDataAndModel` module of the `p2p` package.

Once the 2D field has has been extracted, BGC-val masks the data outside the regions requested in the `regions` option. In this example, that is the `Global` and the `SouthernOcean` regions. These two regions are defined in the `regions` package in the `makeMask` module. This process is described below in sect. 4.3.2.

The `dimensions` option tells BGC-val what the dimensionality of the variable will be after it is loaded, but before it is masked or sliced. The dimensionality of the loaded variable affects how the final results are plotted. For instance, one dimensional variables such as the global total primary production or the total northern hemisphere ice extent can not be plotted with a depth profile, or with a spatial component. Similarly, two dimensional variables such as the air sea flux of $CO_2$ or the mixed layer depth shouldn't be plotted as a depth profile, but can be plotted with percentiles distribution. Three dimensional variables such as the temperature and salinity fields, the nutrient concentrations, and the biogeochemical advected tracers are plotted with time series, depth profile, and percentile distributions. If any specific types of plots are possible but not wanted, they can be switched off using one of the following options:

```
makeTS          : True
makeProfiles    : False
makeP2P         : True
```





The `makeTS` options controls the time series plots, the `makeProfiles` options controls the profile plots, and the `makeP2P` options controls the point to point evaluation plots. These options can be set for each Active Keys section, or they can be set in the global section, described below.

In the case the HadGEM2-ES's chlorophyll section, shown in this example, the absence of an observational data file means
that some evaluation figures will have blank areas, and others figures will not be made at all. For instance, it's impossible to produce a point to point comparison plot without both model and observational data files. The evaluation of `[Chlorophyll]` could be expanded by mirroring the model's coordinate and convert fields with a similar set of data coordinates and convert functions for an observational dataset.

### 4.1.3   Global section

The `[Global]` section of the configuration file can be used to set default behaviour which is common to many evaluation sections. This is because the evaluation sections of the configuration file often use the same option and values in several sections. As an example, the names that a model uses for its coordinates are typically the same between fields; i.e. a Chlorophyll data file will use the same name for the latitude coordinate as the Nitrate data file from the same model. Setting default analysis settings in the `[Global]` section ensures that they don't have to be repeated in each evaluation section. As an example, the
following is a small part of a global settings section:

```
[Global]
model           : ModelName
model_lat       : Latitude
```

These values are now the defaults, and individual evaluation sections of this configuration file no longer require the `model` or
`model_lat` options. However, note that local settings override the global settings. Note that certain options such as `name` or `units` can not be set to a default value.

The global section also includes some options that are not present in the individual field sections. For instance, each configuration file can only produce a single output report, so all the configuration details regarding the html report are kept in the global section:

```
[Global]
makeComp        : True
makeReport      : True
reportdir       : reports/HadGEM2-ES_chl
```

where the `makeComp` is a Boolean flag to turn on the comparison of multiple jobs, models or scenarios. The `makeReport`
is a Boolean flag which turns on the global report making and `reportdir` is the path for the html report.

The global options `jobID`, `year`, `model` and `scenario` can be set to a single value, or can be set to multiple values (separated by a space character), by swapping them with the options: `jobIDs`, `years`, `models` or `scenarios`. For instance, if multiple models were requested, then swap:

```
[Global]
model           : ModelName1
```

with the following:

```
[Global]
models          : ModelName1 ModelName2
```

For the sake of the clarity of the final report, we recommend only setting one of these options with multiple values at one time. The comparison reports are clearest when grouped according to a single setting ie, please don't try to compare too many different models, scenarios, and jobIDs at the same time.

The `[Global]` section also holds the paths to the location on disk where the processed data files and the output im-
ages are to be saved. The images are saved to the paths set with the following global options: `images_ts`, `images_pro`, `images_p2p`, `images_comp` for the timeseries, profiles, point to point and comparisons figures, respectively. Similarly, the post processed data files are saved to the paths set with the following global options: `postproc_ts`, `postproc_pro`, `postproc_p2p` for the timeseries, profiles, and point to point processed data files respectively.

As described above, the global fields `jobID`, `year`, `model` and `scenario` can be used as placeholders in file paths.
Following the bash shell grammar, the placeholders are marked as all capitals with a leading $ sign. For instance, the output directory for the time series images could be set to:

```
[Global]
images_ts : images/$MODEL/$NAME
```

where `$MODEL` and `$NAME` are placeholders for the model name string and the name of the field being evaluated. In the
example in sect. 4.1.2 above, the `images_ts` path would become: `images/HadGEM2-ES/Chlorophyll`. Similarly, the `basedir_model` and `basedir_obs` global options can be used as fill the placeholders `$BASEDIR_MODEL` and `$BASEDIR_OBS` such that the base directory for models or observational data don't need to be repeated in every section.

A full list of the contents of a global section can be found in the `README.md` file. Also, several example configuration files are available in the `ini`.

**4.2  Primary python packages**

In this section, we describe the important packages, that are shown in green in fig. 2. The `timeseries` package is described in sect. 4.2.1 the `p2p` package is described in sect. 4.2.2 the `html` package is described in sect. 4.2.3, and the `bgcvaltools` package is described in sect. 4.2.4. All the figures in sect. 4.2 were produced on the JASMIN computational resource, using the example configuration file `ini/HadGEM2-ES_no3_cmip5_jasmin.ini`, and the html summary report associated with
that configuration file is available in the supplementary materials.



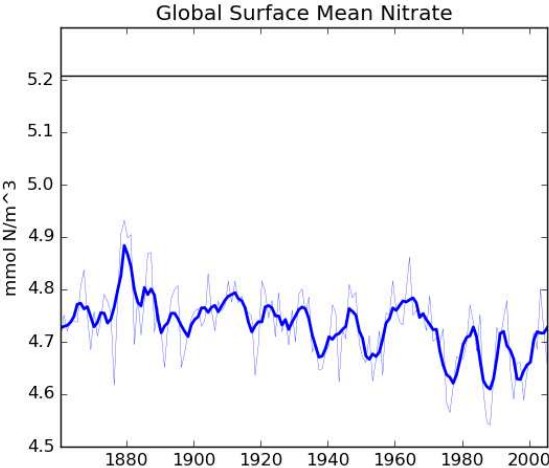

**Figure 3.** A plot produced by the time series package. This figure shows the time development of a single metric, in this case the global surface mean nitrate in HadGEM2-ES in the historical simulation. It also shows the five year moving average of the metric.

### 4.2.1 Time series tools

This `timeseries` package is a set of python tools that produces figures showing the time development of the model. These tools manage the extraction of data from NetCDF files, the calculation of a range of metrics or indices, the storing and loading of of processed data, and the production of figures illustrating these metrics.

5     Firstly, the time development of any combination of depth layer and region can be investigated with these tools. The spatial regions can be taken either from the predefined list or a custom region can be created. The predefined regions are listed in the `regions` directory of the BGC-val. Many metrics are available including, mean, median, minimum, maximum, and all percentiles divisible by 10 (10[th] percentile, 20[th] percentile, etc...). Furthermore, any user defined custom function can also be included as a custom function, for instance the calculation of global total integrated primary production, or the total flux

10  through the Drake Passage.

    The time series tools produce three types of analysis plots. Examples of these three types of figures are shown in figs 3, 4 and 5. All three examples use annual averages of the Nitrate (CMIP5 name: no3) in the surface layer of the global ocean in the HadGEM2-ES model in the historical scenario, in the ensemble member, r1i1p1.

    Figure 3 shows the time development of a single variable: the mean of the nitrate in the surface layer over the entire global

15  ocean, in the HadGEM2-ES model in the historical scenario, in the ensemble member, r1i1p1. This figure shows the annual mean of the HadGEM2-ES model's nitrate as a thin blue line, the five year moving average of of the HadGEM2-ES model's nitrate as a thick blue line, and the World Ocean Atlas (WOA) data (Garcia et al., 2013), shown as a flat black line. The WOA data used here is a climatological dataset, and hence does have a time component. This figure highlights that model simulates a decrease in the mean surface nitrate over the course of the 20[th] century.



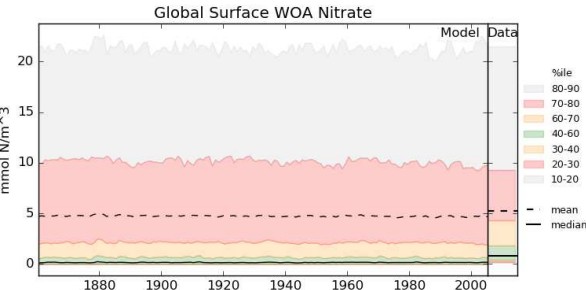

**Figure 4.** A plot produced by the time series package. The figure shows the time developers of many metrics at one: the mean, median and several percentile ranges of the observational data and the model data. In this case, the model data is the global surface mean nitrate in HadGEM2-ES in the historical simulation.

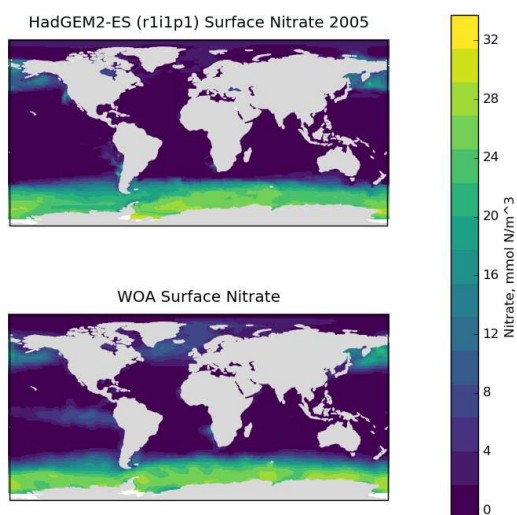

**Figure 5.** A plot produced by the time series package. The figure shows the spatial distribution of the model (top) and the observational dataset (bottom). In this case, the model data is the global surface mean nitrate in HadGEM2-ES in the historical simulation.

Figure 4 shows an example of a percentile range plot, which shows the time development of the spatial distribution of the model data, including the mean, median, and coloured bands to indicate the 10-20, 20-30, 40-60, 60-70 and 70-80 percentile bands. This kind of plot also shows percentile distribution of the spatial distribution of the observational data, in a column on the right hand side. Figure 4 shows the behaviour of nitrate in the surface layer over the entire global ocean, in the HadGEM2-ES model in the historical scenario, in the ensemble member, r1i1p1. This type of plot is produced when the data has for 2 or 3 dimensions but can not be produced for one dimensional model datasets. The percentiles figure can be produced for any layer

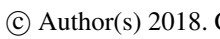


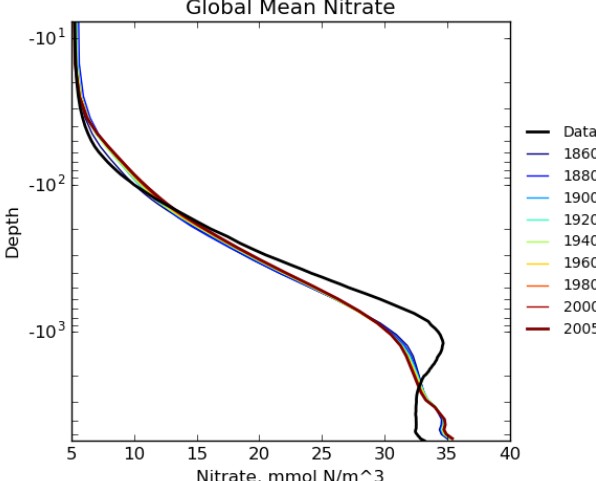

**Figure 6.** The time development of the Global mean dissolved Nitrate over a range of depths. This figure shows the HadGEM2-ES global mean nitrate over the entire water column, each model year is included as a coloured line, and the annual mean of the world ocean atlas nitrate climatology dataset is shown as a black line.

and spatial region and these metrics are all area-weighted. For all three kinds of time series figures, a real dataset can be added, although it is not possible to include the time development of the observational dataset at this stage.

The time series package also produces a figure showing the spatial distribution of the model and observational data. Figure 5 shows an example of such a figure, where the top pane shows the spatial distribution of the final time step of the model, and the bottom pane shows the spatial distribution of the observational dataset. It is possible to plot data for any layer for any region. These spatial distributions are made using the Plate Carré projection, and the projections are set to focus on the region in question. Figure 5 highlights that HadGEM2-ES model failed to capture the high nitrate seen in the observational data in the equatorial pacific.

BGC-val can also produce several figures showing the time development of the model datasets over their entire water columns. The profile modules are stored in the `timeseries` package, as the time series and profiles figures share many of the same underlying methods. The figs 6, 7 and 8 are examples of three profiles plots showing the time development over the water column of the Global mean nitrate in the HadGEM2-ES model in the historical simulation, in the ensemble member, r1i1p1. These plots can only be produced when the data has 3 dimensions. These plots can be made for any region from the predefined list or for custom regions.

Figure 6 shows the time development of the depth profile of the model and observational data. The x-axis shows the value, in this case, the Nitrate concentration in mmol N m$^{-3}$ and the y-axis shows depths. These type of plots always show the first and last time slice of the model, then a subset of the other years are also shown. Each year is assigned a different colour, with



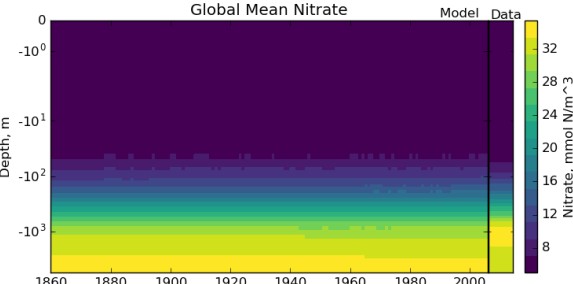

**Figure 7.** The time development of the Global mean dissolved Nitrate over a range of depths. The figure shows the same data as fig. 6, but as a Hovmöller time series plot. The annual mean of the world ocean atlas nitrate climatology dataset is shown as a column on the right hand side of the figure.

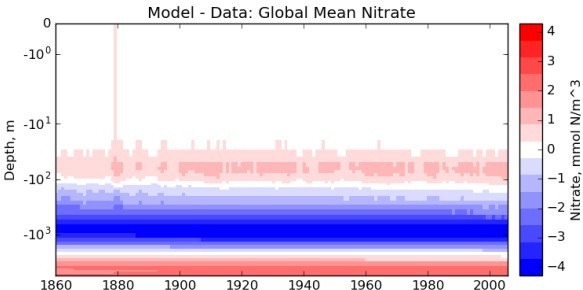

**Figure 8.** The time development of the Global mean dissolved Nitrate over a range of depths. The figure shows the same data as fig. 6 and 7, but with the world ocean atlas nitrate observational measurement subtracted from the model time series.

the colour scale shown in the right hand side legend. If available, the observational data is shown as a black line. This figure shows the annual mean of the world ocean atlas nitrate climatology dataset as a black line.

Figures 7 and 8 are both Hovmöller diagrams (Hovmöller, 1949) and they show the depth profile over time for model and observational data. Figure 7 shows the model and the observational data side by side, and fig. 8 shows the difference between 5 the model data and the observational data. The difference Hovmöller diagrams are only made when the observation dataset is supplied. There appears to be a peak in the difference between the model and the observations over the entire water column in the year 1880. This doesn't appear to be a fault in the model, but simply brief period where the difference was slightly larger than zero. This peak in the mean is also visible in the global mean surface nitrate in fig. 3, but is not visible in the percentile distribution of the surface nitrate in fig. 4.

10 Figures 6, 7 and 8 all show that the model data matches the observational nitrate near the surface, but diverges at depth. The model underestimate the peak in the global mean of the observational nitrate at a depth of approximately 1000m, and then overestimates the observed nitrate below 2000m. Also, the model does not show much interannual variability in the structure





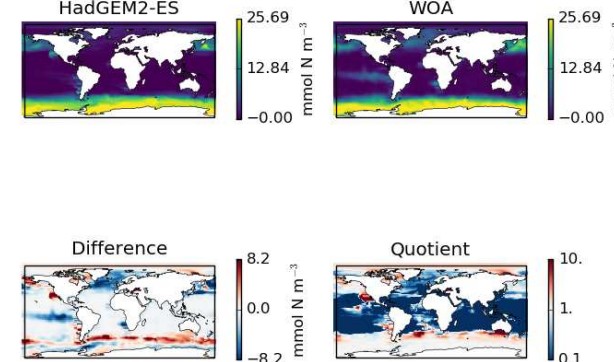

**Figure 9.** Four spatial distributions showing the model (top left), the observational data (top right), and the difference between them (bottom left) and their quotient (bottom right). The model data in these plots is the global surface nitrate in HadGEM2-ES in the historical simulation in ensemble member r1i1p1 in the year 2000. The observational data comes from the annual nitrate dataset in the World Ocean Atlas.

of the global annual average nitrate over the 145 year simulation. However, it is unclear from the WOA annual average whether we should expect any variability from the model over this time range.

In the `timeseries` package, there is also a set of tools for comparing the time series development between multiple versions of the same metric. This is effectively the same as plotting several versions of fig. 3 on the same axes. This kind of diagram can be useful to compare the same measurement between different regions, depth layers, or different members of a model's ensemble. However, these plots can also been used to compare the multiple models. Several examples figure are included in sect. 5, below.

#### 4.2.2 Point to point model data comparison tools

In addition to the time series evaluation, BGC-val can perform for a direct point to point comparison of model against data. The point to point tools here are based on the work by the same author de Mora et al. (2013). In that work, we demonstrated that using point to point analysis is more representative of a real marine dataset than comparing the bulk mean of the model to the bulk mean of the data. The method involves matching the model data to the closest corresponding and observational measurement, then hiding all model points which do not have a corresponding observational measurement and vice versa. The point to point methodology means that the model and observational data have not be re-interpolated to a common grid: They both retain their original grid description.

Figures 9, 10 and 11 show examples of the figures made by the point to point package. In all three figures, the model data is the global surface nitrate in HadGEM2-ES in the historical simulation in ensemble member r1i1p1 in the year 2000. The observational data comes from the nitrate dataset in the World Ocean Atlas Garcia et al. (2013).

Figure 9 is a group of four spatial distributions comparing the model and observational data sets. The top left map is the model, the top right is the observations, the bottom left is the difference between the model and observations (model minus





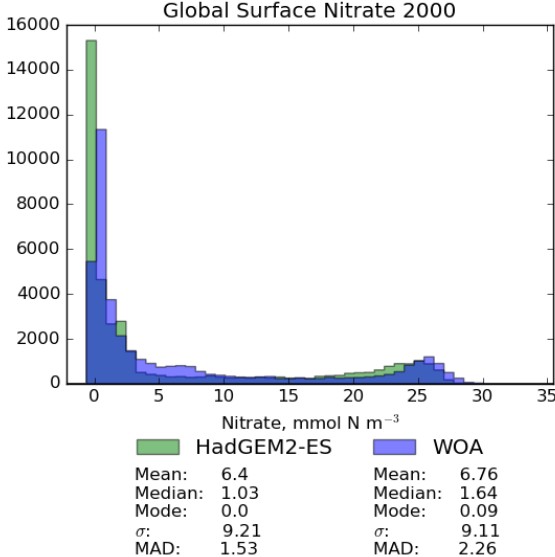

**Figure 10.** A pair of histograms showing the model (green) and the observational data (blue), as well as some metrics of the distribution shape. The model data is the global surface nitrate in HadGEM2-ES in the historical simulation in the year 2000. The observational data comes from the nitrate dataset in the World Ocean Atlas. The metrics are the mean, median, the mode, the standard deviation, $\sigma$, and the Median Absolute Deviation (MAD).

observational data), and bottom right pane is the quotient, (model over observational data). This example shows the comparison at the ocean surface, but these tools also allow a longitudinal or latitudinal transect comparisons or a spatial distribution along specific depth level. This figure shows that the year 2000 of the HadGEM2-ES model reproduce the large scale spatial patterns seen in the observational dataset. The model has significantly higher nitrate that the observational climatology in the Southern

Ocean, the North Pacific and the equatorial regions, and has a significantly lower nitrate in the Arctic regions. A discrepancy in the spatial extent of the high nitrate in the Southern Ocean is shown clearly in the difference pane of this figure. The quotient pane of this figure also shows that model underestimates the low nitrate regions around the tropical waters.

Figure 10 is a pair of histogram showing the same model and the observational data as in fig. 9. This figure also shows some measures of the central tendency, (mean, median, mode), and measures of the deviation (standard deviation and Median

Absolute Deviation) for both model and data. These histograms confirm that the model underestimates the nitrate concentration in the low nitrate region which covers a significant region and is the mode of the WOA dataset.

Figure 11 shows the distribution of the model and the observational data with the model data along the x-axis and the observation data along the y-axis. The 1:1 line is shown as a dashed line; the model overestimates the observation to the right of this line and underestimates to the left of this line. A linear regression is shown as a full line, with the slope, intersect, P-value,

Correlation, and number of data points shown on the right hand side of the figure. In this example, the linear regression is very



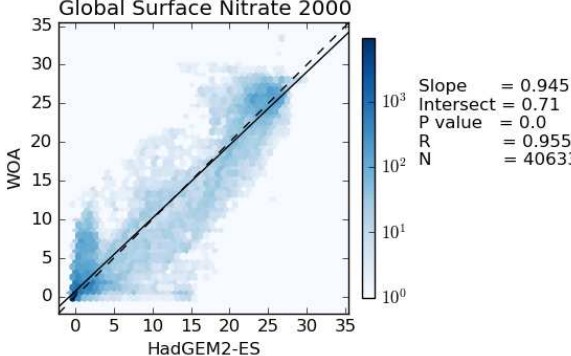

**Figure 11.** This figure shows the distribution of the model and the observational data with the model data along the x-axis and the observation data along the y-axis. The model data is the global surface nitrate in HadGEM2-ES in the historical simulation in the year 2000. The observational data comes from the nitrate dataset in the World Ocean Atlas. The 1:1 line is shown as a dashed line. A linear regression is shown as a full line, with the slope, intersect, P-value, Correlation, and number of data points shown on the right hand side.

close to the 1:1 line, and the bulk of the data is close to a good fit. While the model reproduces the distribution of observational data at low values and high nitrate concentrations, the model overestimates more than half of the nitrate observations between 10 and 20 mmol N m$^{-3}$.

### 4.2.3   HTML report

The `html` package of the BGC-val toolkit contains all the tools needed to produce an report summarising the output of the time series, the profile and the point to point packages. The principal file in this package is the `makeReportConfig` module, which produces an html document according to the settings of the configuration file. Using the configuration file, the report maker finds all the images, and uses several template files to stitch together the individual sections of the report. The html is based on a template taken from https://html5up.net/, used under the Creative Commons Attribution 3.0 License.

An example of the HTML report is available in the supplemental information. This report shows the output of the example configuration file: `ini/HadGEM2-ES_no3_cmip5_jasmin.ini`. In order to access this report, please download and unzip the files, then export them to a local copy before opening the `index.html` file in a browser of your choice.

### 4.2.4   BGC-val tools

This `bgcvaltools` package contains many python routines that perform a range of important functions. These tools include:

a tool to read Netcdf files `dataset.py`, a tool to extract a specific 2D layer or transect `extractLayer.py`, a tool to read





and understand the configuration file, `configparser.py`. There is a wide and diverse selection of tools in this directory: some of them are used regularly by the toolkit, and some are only used in specific circumstances. More details are available in the `README.md` file, and each individual module in the `bgcvaltools` is sufficiently documented that it's role in the toolkit is clear.

## 4.3 User configurable python packages

In this section, we look at the code behind the extensive customiseability of BGC-val: the `functions`, `regions` and `longnames` packages shown in blue in fig. 2 are described in this section. The `functions` package is described in sect. 4.3.1 the `regions` package is described in sect. 4.3.2 the `longnames` package is described in sect. 4.3.3, and the `bgcvaltools` package is described in sect. 4.2.4.

### 4.3.1 Functions

The functions package is a significant contributor of the flexibility of BGC-val. This package allows any operation to be applied to a dataset as the data is loaded. In most cases, the conversion is one the standard functions such as multiply or divide by some arbitrary number, add a constant value to the variable, or simply just load the data as is with no conversion. However, this package can also be used to perform complex data processing.

The `data_convert` and `model_convert` options in the configuration file allow BGC-val to determine which function to apply to the model or observational data as it is loaded. There is no default function, so to simply load the data it is in the file, the standard function `NoChange` should be specified in the `data_convert` or `model_convert` options.

As an example of the structure of a basic function, we look at a simplified version of the `multiplyBy` function, in the `stdfunctions` module of the `functions` package.

```
def multiplyBy(nc,keys, **kwargs):
    f = float(kwargs['factor'])
    return nc.variables[keys[0]][:]*f
```

After declaring the function name and arguments in the first line, this function loads the `factor` from the key word arguments (`kwargs`), and parses it into the single precision floating format in the second line. In the third line, this function loads the

first item in the `keys` list from the NetCDF dataset `nc`, then multiplies that data by the factor `f`, and returns the product. The path to the NetCDF file, the choice of function, the list of keys, and the factor are all provided by the configuration file.

All functions need to be called with the same arguments: `nc` is an NetCDF file opened by the `dataset` module from the `bgcvaltools` package. The `keys` argument is a list of strings which represent the names of fields in a NetCDF file, and the optional `kwargs` argument is used to pass any extra information that is needed (such as a factor or addend).

The key word arguments which are passed to the function must be preceded by the text `model_convert_` or `data_convert_` strings in the configuration file. In the example above, the 'factor' was written in the configuration file as:

```
model_convert_factor : 1e6
```



but it was loaded in the `multiplyBy` function as `kwargs['factor']`.

Some evaluation metrics require multiple variables to be loaded at once and combined together. The `stdfunctions` module of the `functions` package contains a few such medium complexity operations, such as 'sum' which returns the sum of the fields in the `keys` list. The `'divide'` function returns the quotient of the first key over the second key from the `keys` list.

More complex functions can be implemented as well, for instance depth integration, global totals, or the flux through a certain cross section. There are several examples of a complex functions in the functions folder. Note that some of these functions can change the dimensionality of the data, and caution needs to be taken to ensure that the `dimensions` option in the configuration file matches the dimensions of the output of this function.

### 4.3.2 Regions

Similarly to the functions package described above, the regions package allows for expanded flexibility in the evaluation of models. The term 'Region' here is a portmanteau for any selection of data based on its coordinates or values. Typically, these are spatial regional cuts, such as "Northern Hemisphere", but the masking is not limited to spatial regions. For instance, the `regions` package can also be used to remove negative values, to remove zero, NaN or `inf` values.

As an example of the structure of a basic regional mask, we look at the `SouthHemisphere` region, in the `makeMask` module of the `regions` package.

```
def SouthHemisphere(
    name,region,
    xt,xz,xy,xx,xd):
  a = np.ma.masked_where(xy>0.,xd)
  return a.mask
```

where the python standard package `numpy` has been imported as `np`. Each regional masking function has access to the following fields: `name`, the name of the data; `region`, the name of the regional cut; `xt`, a one-dimensional array of the dataset times; `xz`, a one-dimensional array of the dataset depths; `xy`, a one-dimensional array of the dataset latitudes; `xx`, a one-dimensional array of the dataset longitudes; and `xd`, a one-dimensional array of the data. The second to last line creates a masked array of the data array which is masked in all the places where the latitude coordinate is greater than zero (ie: the Northern Hemisphere). The final line returns the mask for this array. All region extraction functions return a numpy mask array. In python, numpy masks are an array of Booleans where True is masked.

Many regions are already defined in the file `regions/makeMask.py`, but it is straightforward to add a new region using the template file, `regions/customMaskTemplate.py`. To do this, make a copy of the `regions/customMaskTemplate.py` file in the `regions` directory, rename the function and file to your mask name, and add whatever cuts are required. BGC-val will be able to locate your region, provided that the region name matches the python function and the region in your configuration file.





### 4.3.3 Long names

In the python source code, objects are often abbreviated or labelled with shorthand, and spaces and hyphens are not acceptable in object names. This means that the internal name of a model, dataset, field, layer, or region is not usually the same in the text that we want to appear in public facing plots. For this reason, the longname package has a dictionary of common terms with

their abbreviated name linked to a 'pretty' name. The dictionary has definitions for each model, scenario, dataset, object, mask, cut, region, field, and other pythonic object used in BGC-val. These pretty names are used when preparing outwards-facing graphics and html pages, such that the name of an object in the configuration file isn't a source of confusion.

This package uses the standard configuration (ini) format for the dictionary. The custom longnames configuration file is simply a long list of short names as the option and long names as the value. for example, the `longnames.ini` includes the

following lines:

```
no3     : Nitrate
chl     : Chlorophyll
```

This means that we can label Nitrate internally as 'no3' as the evaluation name in our configuration file, but when it appears in plots, it will be shown as "Nitrate". Also note that the options are not case sensitive, but the values are case sensitive. While the

default longname list is already relatively extensive, users can add their own longnames to the `longnames/customLongNames.ini` file.

## 5   Applying BGC-val to CMIP5 RCP 8.5

In this section, we show some example figures of the intercomparison of several CMIP5 models. These examples were produced using CMIP5 data on the JASMIN data processing facility, and the configuration file used to produce these is supplied in the

BGC-val git repository under the name `cmip5_rcp85_jasmin.ini` in the `ini` directory. The examples that we show here are the Atlantic meridional overturning circulation in fig. 12, the Antarctic circumpolar current in fig. 13, the total annual air to sea flux of $CO_2$ in fig. 14, the total annual marine primary production in fig. 15, and the global mean surface chlorophyll in fig. 16.

These examples compare a subset of the CMIP5 models in the historical time range and RCP 8.5 scenario in the ensemble

member, r1i1p1. The historical and RCP 8.5 simulations are aligned such that historical simulation links to the RCP scenario at the year 2005. This was done using the `jasmin_cmip5_linking.py` module in the `bgcvaltools` package.

The CMIP5 models shown in these figures are: CESM1-BGC, CMCC-CESM, GFDL-ESM2G, GFDL-ESM2M, GISS-E2-H-CC, GISS-E2-R-CC, HadGEM2-CC, HadGEM2-ES, IPSL-CM5A-MR, IPSL-CM5B-LR, MPI-ESM-LR, MPI-ESM-MR and NorESM1-ME. This report does not include all CMIP5 models, but rather a small number of examples of marine circulation

and biogeochemistry metrics over the historical and RCP 8.5 scenarios. The selection criteria was that the model was required to have biogeochemical datasets in the JASMIN mirror of the CMIP5 data, the r1i1p1 job identifier, and the 'latest' model run





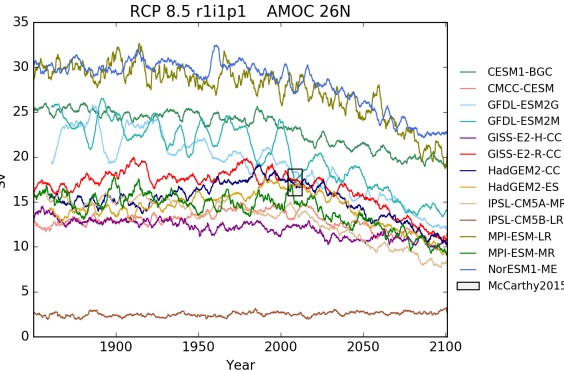

**Figure 12.** The Atlantic Meridianal overturning circulation at 26 North in a subset of CMIP5 models. Each model is shown as a full line, and the historical measurement is shown as a grey area. The model data is a 5 year moving average.

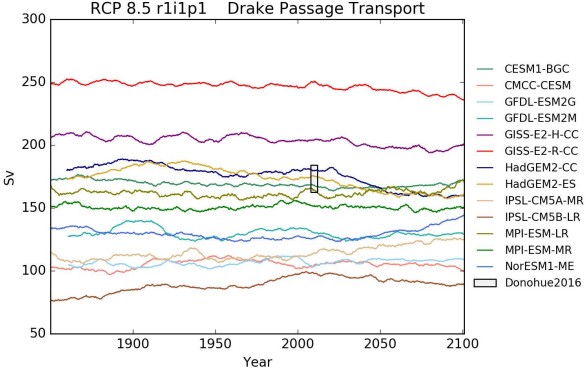

**Figure 13.** The Drake passage current Each model is shown as a full line, and the historical measurement is shown as a grey area. The model data is a 5 year moving average.

tag. Using these tools, we uncovered a previously undetected error in the HadGEM2-ES RCP8.5 r1i1p1 simulation. For these figures, we substituted the r2i1p1 simulation in place of the HadGEM2-ES r1i1p1 simulation.

All five figures here show the 5 year moving average instead of the monthly or annual time resolution of the field in order to improve clarity. The five year window moving average is calculated using the mean of 2.5 years on either side of a central 5 point. This means that the start and end points of the time series are the mean of only 2.5 years.

Table 1 shows the observational measurement the multi-model mean of the years 1975-2000 in the historical scenario, the multi-model mean of the years 2075-2100 under the RCP 8.5 scenario, and the percentage change between the 2075-2100 and 1975-2000 for all five fields.

The Atlantic Meridianal overturning circulation (AMOC) is a major current and consists of two parts: a northbound trans-10 port between the surface and approximately 1200m, and a southbound transport between approximately 1200m and 3000m



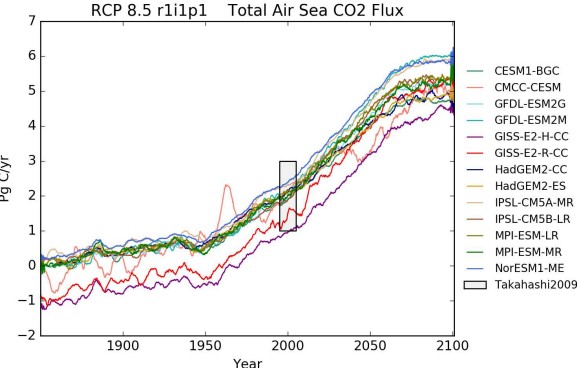

**Figure 14.** The total annual flux of $CO_2$ from the air to the sea in Pg/Year under the RCP8.5 scenario

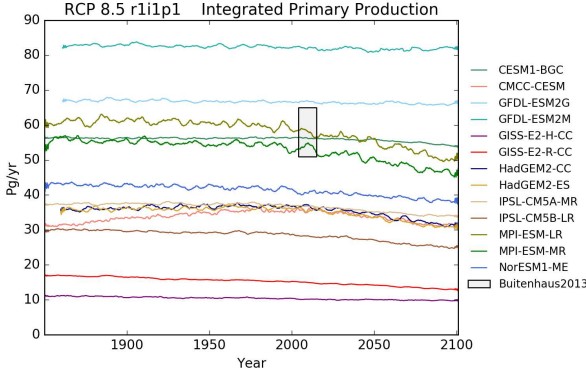

**Figure 15.** The total annual marine primary production of a range of models.

(Kuhlbrodt et al., 2007). The AMOC is responsible for most of the production of roughly half the oceans deep waters (Broecker, 1991). The northward heat transport of the AMOC is substantial, and has a significant role in the Climate of the Northern Hemisphere. The strength of the Northbound AMOC in several CMIP5 models was shown in figure 12.35 of the IPCC report, (Collins et al., 2013). The BGC-val toolkit was able to reproduce the AMOC analyses of the IPCC. As in the IPCC figure,

5  fig. 12 shows the historical and RCP 8.5 projections of the AMOC produced by BGC-val. Please note that we use a different subset of CMIP5 models in this figure relative to IPCCs figure 12.35. The RAPID-array measured the long term mean of the AMOC to be $17.2 \pm 1.5$ Sv between 2004 and 2013(McCarthy et al., 2015). This figure is shown as a black rectangle with a grey face in fig. 12. The calculation was initially based on the methods used by the Met Office's internal ocean evaluation toolkit, Ocean Assess, but was expanded to be model and grid independent. This cross sectional area for the 26° N transect

10  was calculated and saved to a NetCDF file using the `meshgridmaker` module in the `bgcvaltools` package. The model specific cross sectional area was used to calculate the maximum of the depth-integrated cross sectional current, in the custom function `cmip5AMOC` in the `circulation` module in the `functions` package. Amongst the CMIP5 models that included



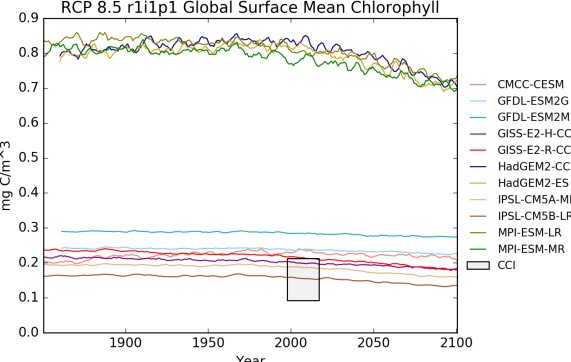

**Figure 16.** The global mean chlorophyll concentration for the surface layer of a range of CMIP5 models.

**Table 1.** Summary table showing the multi model mean and standard deviation of the five fields. After the field and units columns, the observational range, measurement uncertainty, and reference are shown. The fifth column shows the the multi-model mean of years 1975-2000 and the standard deviation ($\sigma$),in the historical simulation. The sixth column shows the the multi-model mean of years 2075-2100 and the standard deviation in the RCP 8.5 simulation. The final column (% Diff.) shows the percentage difference between the first period and the second period.

| Field | Units | Observation | Reference | 1975-2000 | 2075-2100 | % Diff. |
|---|---|---|---|---|---|---|
| AMOC at 26 N | Sv | $17.2 \pm 1.5$ | McCarthy et al. (2015) | $18.0, \sigma: 6.8$ | $13.0, \sigma: 5.3$ | -26 % |
| Drake Passage Transport | Sv | $173.3 \pm 10.6$ | Donohue et al. (2016) | $151, \sigma: 42$ | $149, \sigma: 39$ | -0.9% |
| Total Air Sea $CO_2$ Flux | Pg C y$^{-1}$ | $2 \pm 1$ | Takahashi et al. (2009) | $1.63, \sigma: 0.36$ | $5.2, \sigma: 0.45$ | +240.% |
| Integrated Primary Production | Pg C y$^{-1}$ | $58 \pm 7$ | Buitenhuis et al. (2013) | $43.2, \sigma: 19$ | $40, \sigma: 19$ | -9.3% |
| Global Surface Chlorophyll | mg Chl m$^{-3}$ | 0.09 - 0.21 | ESA Ocean Colour CCI | $0.44, \sigma: 0.29$ | $0.39, \sigma: 0.25$ | -11% |

a Biogeochemical component, several models overestimated the AMOC, and several underestimated the AMOC in the historical simulation. However, nearly all simulations predict a decline in the AMOC over the 21$^{st}$ century, and the multi-model mean drops by 26% from 18 Sv in the mean of the years 1975-2000 to 13 Sv in the mean of the years 2075-2100 under the RCP 8.5 scenario.

5    The Antarctic Circumpolar Current (ACC) is a major current which has a significant impact on the climate of the Southern Ocean and Antarctica. The ACC flows eastward around Antarctica and is the dominant feature of the circulation of the Southern Ocean. The ACC was recently measured through Drake Passage at 173.3±10.7 Sv (Donohue et al., 2016), making the ACC the strongest ocean current in the world. A metric to describe the ACC is the total volume transport through the narrow gap between South America and Antarica, known as the Drake Passage, shown in fig. 13. Here, the Drake passage current is calculated as the

10    total depth-integrated current between the South American coast and the Antarctic peninsula along a line of constant longitude at 78° West. To perform this calculation in a grid independent way, a North-South line was drawn along 78° West, through each





model grid cell there. Like the calculation of the AMOC, above, this calculated was initially based on the methods used by the Met Office's internal ocean evaluation toolkit, Ocean Assess, but was expanded to be model and grid independent. The length of the intersecting line between this line and each grid cell along the line was calculated, then multiplied by the thickness of the layer and the East bound current. These products were summed together to produce the Drake Passage current shown in fig. 13.

This cross sectional area is calculated and saved to a NetCDF file using the `meshgridmaker` module in the `bgcvaltools` package. The calculation was performed in the custom function `cmip5DrakePassage` in the `circulation` module in the `functions` package. Figure 13 shows a moving average with a five year window for several CMIP5 models between the years 1860 and 2100 in units of Sverdrups, and the observation of $173.3 \pm 10.7$ Sv from Donohue et al. (2016). Several CMIP5 models make estimates of the Drake passage transport within the uncertainty of the observational measurement. The

percentage difference between the multi-model means of 1975-2000 and 2075-2100 under the RCP 8.5 scenario is a decrease of 0.9%, even though the inter-model spread is particularly large (70-250 Sv).

The ocean is a major sink of $CO_2$, and absorbed approximately 27% of anthropogenic $CO_2$ emissions between 2002 and 2011 (Le Quéré et al., 2013). The total air sea flux of $CO_2$ from the atmosphere to the ocean is an important metric for understanding the fate of greenhouse gases, (Takahashi et al., 1997). The total global air to sea flux of $CO_2$ from various CMIP5 models is

shown in fig. 14 and the observational range of $2 \pm 1$ Pg C y$^{-1}$ for the year 2000 is taken from Takahashi et al. (2009). Note that the observational data was recorded between 1970 and 2007, but scaled to the year 2000. The calculation was performed in the custom function `TotalAirSeaFluxCO2` in the `AirSeaFluxCO2` module in the `functions` package. The historical period shows a rise in the absorption of $CO_2$ between 1860 and 2005, and that trend is projected to continue into the future under RCP 8.5 scenario. The multi-model mean annual for the years 1975-2000 was 1.63 Pg of Carbon per year, but rose by

240% up to 5.2 Pg of Carbon per year for the years 2075-2100.

The integrated primary production is the global sum of the primary production in the ocean. Marine phytoplankton are responsible for $56 \pm 7$ Pg of primary production per year (Buitenhuis et al., 2013), of similar magnitude to that of land plants (Field et al., 2011). The total primary production is an indicator of the strength of the base of the food chain. Changes in primary production may indicate severe impacts of climate change (Chavez et al., 2011; Anav et al., 2013). In order to calculate this

value, we multiply the primary production from each grid cell by the volume of that grid cell, then take the global sum over the entire ocean. The calculation was performed in the custom function `TotalIntPP` in the `TotalIntPP` module in the `functions` package. Figure 15 shows a wide range of behaviours for the CMIP5 models. Some models show a relatively consistent marine primary production, some models show a constant historical primary production, followed by a decrease in primary production going into the 21st century. One model, CMCC-CESM, even shows an increase in the 20th followed by a

decrease in the 21st century. The multi-model mean annual primary production for the years 1975-2000 was 43.2 Pg of Carbon per year, but decreased by 9% down to 39.7 Pg of Carbon per year for the years 2075-2100.

The concentration of chlorophyll in the surface of the ocean is a indicator of quantity of phytoplankton in the waters. The global mean surface chlorophyll the CMIP5 models (excluding the CESM1-BGC and NorESM1-ME models) is shown in fig. 16. The observational data shown in fig. 16 and in the chlorophyll row of tab. 1 is from the ESA Ocean Colour Climate

Change Initiative (CCI). The model value for global mean surface chlorophyll was calculated taking the five year moving



average of the time series of the area weighted mean of the surface layer for each CMIP5 model. To convert the model data into mg Chl m$^{-3}$, we used the standard function `multiplyBy` with a multiplicand of 1e6, from the `stdfunctions` module in the `functions` package. The CCI global mean surface chlorophyll ranges from 0.09 to 0.21 mg Chl m$^{-3}$ for the years 1997 to 2017. Note that this value represents the range of monthly means and was extracted using the ocean colour data

portal https://www.oceancolour.org/portal/. This value is taken from remote sensing satellites measurements and does not have consistent coverage due to cloud cover and low light in the winter in the polar regions. There appears to be two modes of behaviour in the chlorophyll of the CMIP5 models. The first grouping seems to overestimate the total chlorophyll and the second group is closer to the observed value from ESA Ocean Colour CCI. We hypothesised that the over estimating group can reproduce the low chlorophyll gyres, but likely have too much chlorophyll in the equatorial and Southern Ocean regions, while

the second group might not capture coastal upwelling, which results in the high coastal chlorophyll values. Figures similar to fig. 5, which show spatial distributions for surface chlorophyll for each of the CMIP5 models, are available in the supplemental summary html report. We invite readers to consult this report and draw their own conclusions. The multi-model area-weighted mean surface chlorophyll for the years 1975-2000 was 0.44 mg Chl per cubic meter, but decreased by 11% down to 0.39.6 mg of Chlorophyll per cubic meter for the years 2075-2100. The CCI global monthly mean surface chlorophyll ranges from 0.09

to 0.21 mg Chl m$^{-3}$ for the years 1997 to 2017.

## 6   Conclusions

The biogeochemical evaluation toolkit, BGC-val, is a model and grid independent toolkit that has been built to evaluate marine biogeochemical models using a simple interface. We have presented the ideas that motivated the development of the BGC-val software framework, introduced the code structure, and shown some applications of the toolkit using model results from

CMIP5.

We hope that we have successfully communicated the power and flexibility of this toolkit for the automation of marine model evaluation. This toolkit has already been deployed operationally to evaluate the spin up phase of the marine component of the UKESM1. In the future, the authors will continue to develop and apply the toolkit outlined in this work. Furthermore, as it is straightforward to add new fields and models comparisons to BGC-val, we intend to continue to use this toolkit to compare

UKESM1 and the other models submitted to CMIP6 against each other and against the CMIP5 models.

In addition, the framework that produces these figures was built to make it straightforward to load, mask, compare model and observations, and share results. There are several potential expansions, for instance it may be interesting to evaluate the production of emergent features in marine biogeochemical models, based on the work of (de Mora et al., 2016).

*Code availability.* The BGC-val toolkit is freely available, and distributed with the Berkeley Software Distribution (BSD) 3 clause license.

A fully functional and documented snapshot of the BGC-val toolkit with an associated DOI address will be permanently available via the Zenodo service, https://doi.org/10.5281/zenodo.1215935, (de Mora et al., 2018).





An up to date version will be available at via our in-house gitlab server. Registration for the PML gitlab service is required at http://www.pml.ac.uk/Mode

The up to date code is available to registered users at: https://gitlab.ecosystem-modelling.pml.ac.uk/BGC-val-users/bgc-val.

**Appendix A:  Installing and Running BGC-val**

Specific and up to date details on how to install, set up and run the code can be found in the `README.md` file in the code

repository. However, in this appendix, we present a bare-bones guide on how to use the BGC-val toolkit.

BGC-val was written to be compatible with python 2.7 and has only been tested in a Linux environment. It requires several standard python packages, including Matplotlib, netCDF4, numpy and scipy. It also requires a small number of non-standard packages, such as the UKMO's cartopy package.

While the BGC-val code is available via the Zenodo service (de Mora et al., 2018), this is a snapshot, and can not be

changed once it is published. The up to date versions of this repository will be available only with the gitlab service. For this reason, we recommend using the gitlab version instead of the Zenodo version. Registration for the gitlab service is required at http://www.pml.ac.uk/Modelling_at_PML/Access_Code. The registration process will create a user account for you, and your account will be added to the BGC-val-users group. Once registered, the repository can be clone using the standard git methodology:

```
git clone git@gitlab.em.pml.ac.uk:
BGC-val-users/bgc-val.git
```

however, note that this address may differ in the future.

Once cloned, the BGC-val repository can be installed using the standard python package installer, pip:

```
pip install --user bgc-val-public
```

This will make the tools available in the user's python working space.

To run the code, we advise users to make a copy of the relevant configuration file in the `ini` directory. Their local copy of the configuration file should then be edited as described in sec. 4.1 to reflect their local evaluation requirements.

The BGC-val toolkit is launched by the command:

```
run.py configuration.ini
```

`run.py` is a simple wrapper that passes the local configuration file as a command line argument to the main script, `analysis_parser.p`

*Competing interests.*  The authors are non aware of any competing interests in the publication of this article.

*Disclaimer.*  The software described in this paper is provided 'as is' without warranty of any kind, either express or implied, including, but not limited to, the implied warranties of fitness for a purpose, or the warranty of non-infringement.





We make no warranty that the software will meet your requirements, the software will be uninterrupted, timely, secure or error-free, the results that may be obtained from the use of the software will be effective, accurate or reliable, the quality of the software will meet your expectations, or that any errors in the software will be corrected.

Software and its documentation could include technical or other mistakes, inaccuracies or typographical errors. The software or docu-
mentation here may be out of date, and the authors make no commitment to update such materials. The authors assume no responsibility for errors or omissions in the software or documentation.

In no event shall the authors be liable to you or any third parties for any special, punitive, incidental, indirect or consequential damages of any kind, or any damages whatsoever, including, without limitation, those resulting from loss of use, data or profits, whether or not the authors have been advised of the possibility of such damages, and on any theory of liability, arising out of or in connection with the use of
this software.

The use of this software is done at your own discretion and risk and with agreement that you will be solely responsible for any damage to your computer system or loss of data that results from such activities. No advice or information, whether oral or written, obtained by you from the authors or from this publication web site shall create any warranty for the software.

*Acknowledgements.* The work for this paper was funded through the National Environmental Research Council (NERC) National Capability
grant to the UK Earth System Modelling project (grant numbers NE/N018036/1 and NE/N017951/1). The MOHC contribution was supported by the Joint DECC/Defra Met Office Hadley Centre Climate Programme (GA01101).

We acknowledge use of the JASMIN data processing facility, a collaborative facility supplied by the Centre for Environmental Data Analysis (CEDA) to support the data analysis requirements of the UK and European climate and Earth system modelling community, and would like to thank the JASMIN team for their support.
We also thank the UKESM1 team and Ocean Assess team of the Met Office at the Hadley Centre.

We acknowledge the use of the Ocean Colour Climate Change Initiative dataset, Version 3.1, European Space Agency, available online at http://www.esa-oceancolour-cci.org.



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
