# Peer review of "BGC-val: a model and grid independent python toolkit to evaluate marine biogeochemical models"

_Geoscientific Model Development, 2018_

## Referee Comment (RC1) · Anonymous Referee #1 · 19 Jun 2018

**general comments**

The manuscript from De Mora and co-authors provides a detailed description of both technical and scientific aspects of a new framework, named BGC-val, to perform routine operations and multi-model analyses of marine physical and biogeochemical quantities. Beside the wide range of functions to deeply "dissect" model data, a key aspect is the standardised approach that makes this tool model and grid independent. BGC-val represents a good step forward to support not only a single model but also the analysis of the broad ensemble of data from the CMIP exercises. The rationale supporting the development of the tool is somehow misleading (see specific comments) and can be better tailored to model evaluation purposes rather than development/application. Moreover, the potential of this tool with respect to the existing ones should be better

framed. Overall, the BGC-val workflow is clearly described and an exhaustive set of examples is provided to illustrate its usage and degree of flexibility offered to the end-user. In particular, the different functionalities of the evaluation framework are thoroughly described and the outcomes are collected in a very user-friendly interface (as from the support material).

**specific comments**

1. The introductory sub-section 1.1 provide a long description on different issues, spanning from the degree of complexity in marine ecosystem models, the computational effort required to analyse ocean biogeochemistry in comparison with land system, and the possible strategies to tune ESM models against observational data. Given that the purpose of the section was to describe the ideas that motivated the development of the BGC-val framework, I would have expected instead a clear review of the existing literature to support the need of a new, flexible framework. I suggest the authors to revise this section by including a description of present state of the art tools in comparison to BGC-val and by remarkably resizing less relevant paragraphs related more to model development than their evaluation. In addition, the section will benefit from a short paragraph focusing on the observational datasets that are routinely used and ingested by BGC-val within the validation of UKESM1. This could provide a good starting point to foster the discussion within the modelling community toward the definition of a common framework also for data usage in comparison exercises.

2. As far that overall considerations of computationally cheap or expensive operations are addressed, I see the need for a technical description of BGC-val usage requirements or at least a description of its computational performance/skills on JASMIN system (used CPUs, memory requirements etc...) to allow the end-user determining beforehand if the tool can be deployed on its own system. I also suggest to revise the Section 3 as single paragraph by streamlining the text on the workflow (which already have a stepwise organization) and by removing redundant comments between the existing subsections. Finally, I guess that BGC-val may include some degree of

parallelism, if so, a brief description of where parallel instances/computations are performed within the workflow could be very helpful.

3. The proposed example of BGC-val summary report contains analysis also for physical ocean quantities, e.g. Drake passage flow and AMOC. Authors report that computations for these quantities were adapted from the "Ocean Assess" tool, which is unfortunately not available to the public and methodologies are not clearly referenced. For such a reason, I think it is preferable to avoid pointing/referencing at "Ocean Assess" methodologies (see section 1.2 and 5). Authors can instead provide some details on the computation methods for these metrics (if relevant to the manuscript) or improve the description of custom functions reported in section 5 (namely cmip5DrakePassage, cmip5AMOC).

4. Authors clearly state that they "will continue to develop and apply the toolkit" in the future which is quite an interesting and promising perspective. However this seems to stride with the development of ESMValTool that is also receiving contributions and support from authors (see section 1.2). At this stage, it seems reasonable to assume that ESMValTool will include in the next period several features from BGC-val. It looks to me that BGC-val will become a mirror of ESMValTool ocean analysis at some point, so could it be possible to converge into a single tool instead of having duplicated efforts?

**technical corrections**

P1.L9 : the expression "marine circulate and biogeochemical parts" isn't really clear, maybe authors meant something like marine physical and biogeochemical quantities.

P2.L20 : the term "sequesterer" might be suitably replaced with "sink"

P2.L28 : remove the comma at " range, (.."

P3.L13-14: revise text as " . . . any component of an Earth system model . . ." and check if commas before citations are always in the correct place.

P7.L9 : this sentence is not totally clear.

[Figure]

P7.L10 : I guess is " . . . the new masks can be defined in advance, . . . "

P8.L10 : it is a detail but actually in the supplementary material there is only one index file of example.

P8.L18 : " . . . stored in one or several .." P9.L9 : please revise as " ...it is much quicker to evaluate a 2D field ..." or "...it is much quicker the evaluation a 2D field ..."

P10.L11: "In contrast, the stage ..."

P10.L22 : "observational dataset for the model to match against. ..."

P18.L4: repeated "of"

P18.L15: i think it is not necessary to repeat here and in the following text the reference to "in the HadGEM2-ES model in the historical scenario, in the ensemble member, r1i1p1" already given in the initial description of figures 3-5.

P24.L14: I think that paragraph 4.2.4 can be easily integrated somewhere else in the text above or in the general description of the package structure. Accordingly modify the reference in P25.L8 ("and the bgcvaltools package is described in sect. 4.2.4.") and other instances.

P25.L12: "the conversion is one of the standard functions "

P27.L9 : Missing capital font at " For example, the ..."

P27.L19 : Please add a reference or a http address describing the JASMIN facility

P27.L31: here it is implicitly assumed that the reader knows about the JASMIN system, maybe some detail at least on the "mirror" could be helpful.

P28.L1-2 : Maybe some more hints on this error can be profitable to describe the effective benefits of routinely use BGC-val for model development and simulations.

P29.L7 : space is missing at " . . . between 2014 and 2013 (McCarthy et al . . . "

P28.L9 & P31.L2 : here authors refers to a methodology from the "Ocean Assess" tool that has no references.

P31.L34 : Please report here the details about ESACCI product, as given in the final notes at P34.L21

P32.L13 : there is a typo in the chla value reported to be "0.39.6" mg chl m-3.

P32.L14 : the sentence on data starting with "The CCI global monthly mean .. " was already quoted at line 3 and should be removed from here.

---

## Referee Comment (RC2) · Anonymous Referee #2 · 4 Jul 2018

**General Comments**

de Mora et al., present a overview, introduction and description of the BGC-val toolkit for evaluating biogeochemical models. It includes the motivating philosophy, the structure and basic use of the toolkit demonstrated by examples. I have not personally used BGC-val before so hopefully my comments are a useful measure of the accessibility of the manuscript to the wider biogeochemical modelling community.

Overall, the toolkit itself has many useful and valuable features including the grid-independence which facilitates straightforward inter-model comparisons without the issues of re-gridding model and observation data. The use of front-loading functions and the html output make this is a very user-friendly toolkit which is also welcoming. The manuscript provides a thorough and detailed description of the toolkit that would

be a useful resource and basic guide for potential users. However, the manuscript is quite long and verbose in places and so would benefit greatly from improvements to its structure and presentation. I have provided specific comments including suggestions on making the manuscript clearer below.

**Specific Comments**

Abstract: It would be useful to state the intention of the manuscript upfront echoing the text on Pg 3 lines 2 - 5

Pg 2, lines 19 - 33: Much of the text here seems to repeat ideas and themes from the preceding part of the introduction which makes the Introduction as a whole difficult to follow. I would suggest the text on UKESM1 follows well from the CMIP text, and the text between be incorporated into the first few paragraphs.

Pg 2, line 20: 'sink' would be a more appropriate term rather than 'sequesterer'

Section 1.1: I appreciate the discussion in this section, as it's rare to find reflections of this kind. The section raises a number of important issues such as the scarcity and uncertainty associated with observations and the trade-off of between model complexity and computational efficiency. However I think these these require a more quantitive approach to model evaluation (e.g., Stow et al., 2009, Progress in Marine Systems; Kriest et al., 2012, Global Biogeochemical Cycles; Buchanan et al., 2018, Global Biogeochemical Cycles) that is not currently available with this toolkit. Therefore, I don't think this section fits well within the manuscript and could be removed to make a more concise manuscript and still be equally strong.

Pg 3, line 27: please briefly elaborate on the influence of biology on physical circulation

Pg 4, lines 8 - 10: export production is a pertinent example here that could be included to provide a biology-specific example e.g., Boyd Trull (2007) Progress in Oceanography; Henson et al., (2011) Geophysical Research Letters

Section 2: Concepts such as grid-cells and masking are defined here in a number of

sentences which seems unnecessary given that readers interested in a biogeochemical model evaluation toolkit are likely to know these concepts. Removing or cutting these sentences down would streamline the text and make it easier to read.

Pg. 6, line 11: this is very similar to the preceding sentences

Pg. 18 line 18: '. . .is a climatological dataset, and hence does have a time component', is a 'not' missing here?

Pg. 22 line 22: I would generally take 'point to point analysis' in a model-data comparison context to refer to the use of individual bottle measurements rather than climatological data. I'm not clear which of these this section is demonstrating. In either case, is there a procedure for when multiple observations correspond with a single model grid-box?

Pg. 32, lines 6 - 15: I think the description of results throughout highlights the use of the toolkit well but I think this text is presenting some extra results with the hypothesis which I don't think is appropriate in this type of manuscript.

Figures: There are a number of figures given as examples of BGC-val output but they are quite difficult to link to the different packages discussed. I think it would be much clearer and easier to comprehend if the figures were grouped as sub-panels in individual figures. For example, Figures 3, 4, 5 could comprise a figure demonstrating the timeseries package, Figures 6,7,8 would demonstrate the timeseries package with a depth component, Figures 9,10,11 would demonstrate figures from the point-to-point package etc. . .

Figures: What control do you have in setting figure characteristics such as the min/max of scales, and colour scale? Are there options to export the figures in different formats, e.g.. bitmap and vector formats?

Section 5: Many of the concepts such as AMOC, ACC, anthropogenic $CO_2$ and productivity discussed are each defined in a number of sentences. Again, it would seem

that readers would be mostly familiar with these concepts. Removing or cutting these sentences down would help streamline the section and make it easier to demonstrate the toolkit capabilities.

Conclusions: Can you expand on potential future developments or expansions? For example, could quantitative model evaluation be built into this toolkit such as Taylor diagrams and other metrics (Jolliff et al., 2009; Stow et al., 2009 in Journal of Marine Systems). Does the design of the code facilitate this?

---

## Author Comment (AC1) · 25 Jul 2018

Firstly, we'd like to thank both reviewers and the editors at GMD for their kind words and valuable comments. The authors feel that this paper has been significantly improved as a result of their efforts.

In the general comments, both reviewers suggested that we reduce the length of the subsection 1.1 and instead replace it with more information about the existing tools. These changes have been made and instead of adding the new text here, we refer you to read the new section in the attached draft.

Both reviewers wanted more information about the future development of this tool and how it relates to ESMValTool. Fortunately, since submitting this paper, Lee de Mora

has become a contributor to the ESMValTool repository and has learned a lot about the current state of that toolkit. In addition to the changes to section 1.1, we have added the following section to the conclusion:

> While it is a separate toolkit, many of the evaluation metrics used in BGC-val will also be ported onto the ESMValTool (Poloczanska et al., 2016) platform by the authors of this paper. When ported into ESMValTool version 2, these metrics will be made available for use by the wider Earth system model evaluation community.

Both reviewers also caught the absence of citations for the met office toolkit, Ocean assess. Unfortunately, the authors are not aware of a reference to describe the ocean assess toolkit, and don't know if there a public facing description of this tool; we can't find a paper, a website, or a publicly visible code repository. However, we hesitate to remove ocean assess from the manuscript as it was a significant progenitor of this toolkit and it is a valuable tool in its own right. We have gone into more detail about Ocean Assess in Section 1.1.

The specific comments from both reviewers have also been addressed in the revised article and the changes are outlined below. The reviewer comments are quoted in italics and ours comments are shown in normal font afterwards. The technical corrections were also addressed in full, but as they were straightforward, we do not repeated them here. Also note that we have changed the DOI to the code on Zenodo to reflect the latest version of BGC-val.

Once again, thanks for your efforts reviewing and editing this manuscript,

Sincerely,

Lee de Mora, Andrew Yool, Julien Palmieri, Alistair Sellar, Till Kuhlbrodt , Ekaterina Popova, Colin Jones, and J. Icarus Allen

**1 Anonymous Referee #1**

**1.1 General comments**

*The manuscript from De Mora and co-authors provides a detailed description of both technical and scientific aspects of a new framework, named BGC-val, to perform routine operations and multi-model analyses of marine physical and biogeochemical quantities. Beside the wide range of functions to deeply dissect model data, a key aspect is the standardised approach that makes this tool model and grid independent. BGC-val represents a good step forward to support not only a single model but also the analysis of the broad ensemble of data from the CMIP exercises. The rationale supporting the development of the tool is somehow misleading (see specific comments) and can be better tailored to model evaluation purposes rather than development/application. Moreover, the potential of this tool with respect to the existing ones should be better framed. Overall, the BGC-val work flow is clearly described and an exhaustive set of examples is provided to illustrate its usage and degree of flexibility offered to the end-user. In particular, the different functionalities of the evaluation framework are thoroughly described and the outcomes are collected in a very user-friendly interface (as from the support material).*

Thank you for their kind words and clear summary.

**1.2 Specific comments**

*1. The introductory sub-section 1.1 provide a long description on different issues, spanning from the degree of complexity in marine ecosystem models, the computa-*

*tional effort required to analyse ocean biogeochemistry in comparison with land system, and the possible strategies to tune ESM models against observational data. Given that the purpose of the section was to describe the ideas that motivated the development of the BGC-val framework, I would have expected instead a clear review of the existing literature to support the need of a new, flexible framework. I suggest the authors to revise this section by including a description of present state of the art tools in comparison to BGC-val and by remarkably re-sizing less relevant paragraphs related more to model development than their evaluation. In addition, the section will benefit from a short paragraph focusing on the observational datasets that are routinely used and ingested by BGC-val within the validation of UKESM1. This could provide a good starting point to foster the discussion within the modelling community toward the definition of a common framework also for data usage in comparison exercises.*

Both reviewers have suggested that we reduce the length of the subsection 1.1 and instead replace it with more information about the existing tools, and information about the observational data used. These changes have been made and instead of adding the new text here, we refer you to the new section in the attached draft.

*2. As far that overall considerations of computationally cheap or expensive operations are addressed, I see the need for a technical description of BGC-val usage requirements or at least a description of its computational performance/skills on JASMIN system (used CPU's, memory requirements etc...) to allow the end-user determining beforehand if the tool can be deployed on its own system. I also suggest to revise the Section 3 as single paragraph by streamlining the text on the work flow (which already have a step-wise organisation) and by removing redundant comments between the existing subsections. Finally, I guess that BGC-val may include some degree of parallelism, if so, a brief description of where parallel instances/computations are performed within the work flow could be very helpful.*

We've added the following paragraph to the end of section 5:

[Figure]

The computation cost required to perform these evaluations depends on several factors: including the number of models being investigated, the number of years being investigated, the size of the model grid, the number of depth fields, the number of metrics requested, the number of regions requested, the number of depth layers requested, the number of fields under investigation, and the power of the computational system being used. To give a coarse estimate of the computational cost of the tool, we applied BGC-val to a single model (HadGEM2-ES), for a single CMIP5 field (no3), over a single layer (surface), in a single region (global), over the entire CMIP5 historical period (1850-2007), and ran the time series, profile and a point to point comparison and the html report maker. We used the JASMIN sci1 processing node, and ran three iterations. The wallclock time needed to run all evaluation metrics, produce all plots, and make the final html report was 5 minutes 39 seconds, as reported by the Linux utility `time`.

*3. The proposed example of BGC-val summary report contains analysis also for physical ocean quantities, e.g. Drake passage flow and AMOC. Authors report that computations for these quantities were adapted from the Ocean Assess tool, which is unfortunately not available to the public and methodologies are not clearly referenced. For such a reason, I think it is preferable to avoid pointing/referencing at Ocean Assess methodologies (see section 1.2 and 5). Authors can instead provide some details on the computation methods for these metrics (if relevant to the manuscript) or improve the description of custom functions reported in section 5 (namely cmip5DrakePassage, cmip5AMOC).*

We made some comments about this in the introduction of this document. Unfortunately, we're not able to find a reference for Ocean Assess.

*4. Authors clearly state that they will continue to develop and apply the toolkit in the*

*future which is quite an interesting and promising perspective. However this seems to stride with the development of ESMValTool that is also receiving contributions and support from authors (see section 1.2). At this stage, it seems reasonable to assume that ESMValTool will include in the next period several features from BGC-val. It looks to me that BGC-val will become a mirror of ESMValTool ocean analysis at some point, so could it be possible to converge into a single tool instead of having duplicated efforts?*

We have added more information about our contributions to ESMValTool in the conclusion section. This was also reproduced in this response letter, above.

**1.3   Technical corrections**

The technical corrections were also addressed in full, but we have not reproduced these here.

**2   Anonymous Referee #2**

**2.1   General Comments**

*de Mora et al., present a overview, introduction and description of the BGC-val toolkit for evaluating biogeochemical models. It includes the motivating philosophy, the structure and basic use of the toolkit demonstrated by examples. I have not personally used BGC-val before so hopefully my comments are a useful measure of the accessibility of the manuscript to the wider biogeochemical modelling community. Overall, the toolkit*

*itself has many useful and valuable features including the grid independence which facilitates straightforward inter-model comparisons without the issues of re-gridding model and observation data. The use of front-loading functions and the html output make this is a very user-friendly toolkit which is also welcoming. The manuscript provides a thorough and detailed description of the toolkit that would be a useful resource and basic guide for potential users. However, the manuscript is quite long and verbose in places and so would benefit greatly from improvements to its structure and presentation. I have provided specific comments including suggestions on making the manuscript clearer below.*

Thank you for their kind words and clear summary.

2.2  Specific Comments

*Abstract: It would be useful to state the intention of the manuscript upfront echoing the text on Pg 3 lines 2-5:*

We added the following lines to the abstract:

> A brief outline of how to access and install the repository is presented in appendix A, but the specific details on how to use the toolkit are available in the code repository.

*Pg 2, lines 19 - 33: Much of the text here seems to repeat ideas and themes from the preceding part of the introduction which makes the Introduction as a whole difficult to follow. I would suggest the text on UKESM1 follows well from the CMIP text, and the text between be incorporated into the first few paragraphs.*

We moved the two paragraphs (p2 lines 25-34) to the start of the introduction chapter.

*Section 1.1: I appreciate the discussion in this section, as itŠs rare to find reflections of this kind. The section raises a number of important issues such as the scarcity and uncertainty associated with observations and the trade-off of between model complexity and computational efficiency. However I think these these require a more quantitative approach to model evaluation (e.g., Stow et al., 2009, Progress in Marine Systems; Kriest et al., 2012, Global Biogeochemical Cycles; Buchanan et al., 2018, Global Biogeochemical Cycles) that is not currently available with this toolkit. Therefore, I don't think this section fits well within the manuscript and could be removed to make a more concise manuscript and still be equally strong.*

This section was removed, and the rest of the subsection was refocused on model evaluation.

*Pg 3, line 27: please briefly elaborate on the influence of biology on physical circulation*

We were referring to the impact of self-shading on water column temperature. However, the section was removed, as suggested by reviewer #1.

*Pg 4, lines 8 - 10: export production is a pertinent example here that could be included to provide a biology-specific example e.g., Boyd Trull (2007) Progress in Oceanography; Henson et al., (2011) Geophysical Research Letters*

Thanks, these references were added as examples.

*Section 2: Concepts such as grid-cells and masking are defined here in a number of sentences which seems unnecessary given that readers interested in a biogeochemical model evaluation toolkit are likely to know these concepts. Removing or cutting these sentences down would streamline the text and make it easier to read.*

[Figure]

These sentences were cut.

*Pg. 6, line 11: this is very similar to the preceding sentences*

Removed entire paragraph.

*Pg. 18 line 18: ... is a climatological dataset, and hence does have a time component, is a 'not' missing here?*

For clarity, we changed this sentence to:

> The WOA data used here is an annual-average climatological dataset, and hence does not have a time component.

*Pg. 22 line 22: I would generally take point to point analysis in a model-data comparison context to refer to the use of individual bottle measurements rather than climatological data. I'm not clear which of these this section is demonstrating. In either case, is there a procedure for when multiple observations correspond with a single model grid-box?*

Point-to-point is a valid description of this case, because even the most thorough biogeochemical datasets typically contain regions with no data. We try to avoid interpolated datasets for this work. Nevertheless, point to point is the best way to compare models to observational dataset without interpolating them to a common grid. The point to point tools are fully described in de Mora 2012 (gmd-6-533-2013). We continue to use the methods described in that work, where we took the mean of all data points when multiple observations correspond with a single model grid box and vice versa.

*Pg. 32, lines 6 - 15: I think the description of results throughout highlights the use of the toolkit well but I think this text is presenting some extra results with the hypothesis which I don't think is appropriate in this type of manuscript.*

This text was removed.

*Figures: There are a number of figures given as examples of BGC-val output but they are quite difficult to link to the different packages discussed. I think it would be much clearer and easier to comprehend if the figures were grouped as sub-panels in individual figures. For example, Figures 3, 4, 5 could comprise a figure demonstrating the time series package, Figures 6, 7, 8 would demonstrate the time series package with a depth component, Figures 9, 10, 11 would demonstrate figures from the point-to-point package etc. . .*

This was initially our plan, but we found that the shape and size of the automatically generated figures did not fit together very cleanly on a common figure. Either one axis became unreasonably small, or we were forced to include large white-space areas. We also feel that the plots shown in the paper should be representative of the automatically generated ones, so we did not want to change the aspect ratio as they currently fit into the html report. For this reason, we were forced to plot the figures as they were here. We hope that it will become clearer to read when the article is published in the two column non-discussion format.

*Figures: What control do you have in setting figure characteristics such as the min/max of scales, and colour scale? Are there options to export the figures in different formats, e.g.. bitmap and vector formats?*

The figures are exported in the raster graphics format, PNG, as this is the easiest format for exporting from the python graphical toolkit that we use (Matplotlib). The colour scales, time scale and min/max range are set automatically by the contents of

the data. Similarly, the latitude and longitude range in map plots is determined by the scope of the data after all regional masks have been applied.

*Section 5: Many of the concepts such as AMOC, ACC, anthropogenic CO2 and pro-ductivity discussed are each defined in a number of sentences. Again, it would seem that readers would be mostly familiar with these concepts. Removing or cutting these sentences down would help streamline the section and make it easier to demonstrate the toolkit capabilities.*

I would argue that defining these terms at the first use is fairly crucial in technical publications like this one. For instance, it is not impossible that a future reader of this article comes to this paper from an atmospheric evaluation perspective (or computer science perspective, for that matter). These ocean-specific terms need to be defined in order that non-ocean scientists can understand their significance.

*Conclusions: Can you expand on potential future developments or expansions? For example, could quantitative model evaluation be built into this toolkit such as Taylor diagrams and other metrics (Jolliff et al., 2009; Stow et al., 2009 in Journal of Marine Systems). Does the design of the code facilitate this?*

Indeed it does, and in fact, we have previously implemented both Taylor and Target diagrams in an older version of this toolkit, and they could be updated to fit the recent version of BGC-val. I have added the following text to the manuscript:

[revised manuscript text omitted]

---

## Author Response (AR3)

To GMD editors and reviewers,

Please find below the version controlled version of GMD-2018-103 which answers the minor revision requested by Julia Hargreaves on the 4th of September 2018.

At the suggestion of the editor, I've actually patched the BGC-val toolkit so that it can produce figures in png or svg format, and remade all the figures in svg format. (I'll also be adding a similar patch into ESMValTool, which I'm sure will benefit many people using that tool in the future.)

The GMD manuscript preparation instructions do not include svg in it's list of acceptable image formats (*.pdf, *.ps, *.eps, *.jpg, *.png, or *.tif ), so I've converted them from svg to pdfs.

With regards to the readme file, it's a standard practice for readme files to be included in source code repositories. I have changed to text to explicitly describe where to find the readme. In the attached document, please find these changes at p3-L4 - p3 L9.

Elsewhere, I have remove the urls of the registration page and instead point readers towards the code availability section (P7 L16, P33 L9).

Thanks for all your help with the review process,

Lee de Mora

[revised manuscript text omitted]